# DoFlow: Flow-based Generative Models for Interventional and Counterfactual Forecasting on Time Series

**Dongze Wu & Yao Xie**
H. Milton Stewart School of Industrial and Systems Engineering
Georgia Institute of Technology
Atlanta, GA 30332, USA
`dwu381@gatech.edu, yao.xie@isye.gatech.edu`

**Feng Qiu**
Argonne National Laboratory
Lemont, IL 60439, USA
`fqiu@anl.gov`

## Abstract

Time-series forecasting increasingly demands not only accurate observational predictions but also causal forecasting under interventional and counterfactual queries in multivariate systems. We present DoFlow, a flow-based generative model defined over a causal Directed Acyclic Graph (DAG) that delivers coherent observational and interventional predictions, as well as counterfactuals through the natural encoding–decoding mechanism of continuous normalizing flows (CNFs). We also provide a supporting counterfactual recovery theory under certain assumptions. Beyond forecasting, DoFlow provides explicit likelihoods of future trajectories, enabling principled anomaly detection. Experiments on synthetic datasets with various causal DAG structures and real-world hydropower and cancer-treatment time series show that DoFlow achieves accurate system-wide observational forecasting, enables causal forecasting over interventional and counterfactual queries, and effectively detects anomalies. This work contributes to the broader goal of unifying causal reasoning and generative modeling for complex dynamical systems.

## 1 Introduction

Forecasting the evolution of multivariate time series is a central problem in statistics and machine learning, with extensive development across both classical and modern approaches. In the standard setting, a model observes past values and predicts what future values are likely to be under the same conditions that generated the past data. These models are purely *observational*: they learn correlations from past behavior and extrapolate them into the future.

However, many real-world applications require more than observational forecasting. Practitioners often need to answer "what if" questions about interventions, i.e., how the system would evolve if certain control variables were set differently. In this work, we focus on two types of causal queries:

- An *interventional query* asks: "How will the forecast change under a planned modification of certain variables?" Here, the goal is to predict the future trajectory under a chosen sequence of actions. For example, in hydropower time series, turbine-control signals are upstream causes of downstream signals such as power output (Klein et al., 2021). Operators may specify a hypothetical turbine-control plan and ask how all downstream signals will evolve. This contrasts with standard observational forecasters, which, given a fixed past context, produce a fixed forecast. They do not allow one to modify future control plans and inspect how changes propagate through the system.
- A *counterfactual query* instead asks: "What would the system trajectories have looked like had we intervened differently?" Here, we have already observed a factual trajectory and ask

how the *same* trajectory would have changed under an alternative intervention. For example, in healthcare (Cheng et al., 2020; Jacob Rodrigues et al., 2020), we may observe a patient's treatment and outcome trajectory, and then ask whether this particular patient's outcome trajectory would have been better or worse under a different dosing schedule. Outcomes depend not only on dosing but also on patient-specific factors that are not directly observed (e.g., baseline health), yet these factors are implicitly reflected in the factual trajectory. By conditioning on the factual trajectory, we can infer such unobserved factors and then re-simulate how this same patient's trajectory would have evolved under the alternative dosing plan.

However, most modern forecasters are *observational*: they excel at correlational prediction but cannot answer causal queries. Also, to our knowledge, there exists no general framework for counterfactual time-series forecasting yet. Bridging these regimes calls for a model that is both *causally structured* and *generative*, capable of generating consistent and system-wide trajectories under causal queries.

**Contributions.** In this paper, we address this gap by proposing *DoFlow*, a generative model based on Continuous Normalizing Flows (CNFs) that explicitly embeds a causal directed acyclic graph (DAG) structure. Leveraging the invertibility of CNFs and the temporal conditioning in Neural ODEs, DoFlow provides a unified framework for *observational*, *interventional*, and *counterfactual* forecasting. Beyond forecasting, it also yields explicit likelihoods for future trajectories, enabling principled anomaly detection. By unifying causal reasoning with deep generative modeling, DoFlow advances toward trustworthy inference and decision support in complex dynamical systems.

## 1.1 RELATED WORK

To situate our work, we briefly review related lines of research spanning time-series forecasting and causal generative modeling.

- *Time-series forecasting.* Modern approaches can be broadly categorized into four families: (i) classical statistical models such as ARIMA, state-space models, and vector autoregression (Young and Shellswell, 1972; Harvey, 1990; Hyndman et al., 2008; Zivot and Wang, 2006; Scott and Varian, 2014; Taylor and Letham, 2018); (ii) deep sequence models including RNNs/LSTMs and attention-augmented variants (Medsker et al., 2001; Graves, 2012; Qin et al., 2017; Lindemann et al., 2021); (iii) Transformer-based architectures offering scalability on long sequences (Vaswani et al., 2017; Zhou et al., 2021; Wu et al., 2021; Nie et al., 2022); and (iv) recent flow and diffusion models that directly generate trajectories, including autoregressive (Rasul et al., 2020; 2021; Gao et al., 2025), graph-augmented (Dai and Chen, 2022), and horizon-wide designs (Alcaraz and Strodthoff, 2022; Shen and Kwok, 2023). These models typically condition on learned context representations and use expressive generative backbones to produce future forecasts, but they remain *observational*, lacking the ability to simulate interventional and counterfactual trajectories.
- *Causal generative modeling.* For static data, interventional and counterfactual queries are traditionally addressed via structural equation models and do-calculus. Recent generative extensions include graph causal encoders (Schölkopf et al., 2021; Sánchez-Martin et al., 2022; Rahman and Kocaoglu, 2024), diffusion and flow-based models (Khemakhem et al., 2021; Javaloy et al., 2023; Chao et al., 2024; Wu et al., 2025), and model-agnostic counterfactual generators (Karimi et al., 2020; Dasgupta et al., 2025). However, these methods largely target *non-temporal* settings and do not capture causal dependencies over time. A related work studies counterfactual dynamic forecasting (Liu et al., 2023) in constrained physical systems, with interventions only on the initial constrained state. By contrast, *DoFlow* extends causal generative modeling to general *time-indexed DAGs*, enabling coherent interventional and counterfactual forecasting across dynamic systems.

We briefly situate DoFlow relative to two complementary lines a practitioner may combine with our method: (i) Causal effect estimation—quantifying how external actions change short-term expected outcomes, and (ii) Causal discovery—recovering a causal DAG from observational data.

- *Causal effects on time series.* Prior research on time-series interventions has primarily focused on estimating treatment effects under external actions, using variational Bayesian models, and classical time-series approaches like ARIMA (Brodersen et al., 2015; Scott and Varian, 2014; Linden, 2015; Bernal et al., 2017; Bica et al., 2020b;a; Menchetti et al., 2021; Wijn et al., 2022; Grecov et al., 2022; Giudice et al., 2022; Runge et al., 2023; Hyndman and Rostami-Tabar, 2025). These studies typically model how a discrete action variable $A_t \in \{0, \ldots, J-1\}$—for example,

representing treatment versus control or different policy levels—affects future outcomes of the series. These are often formalized as $\tau_t = \mathbb{E}[Y_t|A_{t-1} = j] - \mathbb{E}[Y_t|A_{t-1} = k]$, where $Y_t$ denotes the outcome and $A_t$ the time-varying action. Methodologically, modern approaches estimate the expected outcome under alternative actions, often through conditional generative or variational representation-learning models, to compute average or individualized treatment effects (Lim, 2018; Liu et al., 2020; Li et al., 2021; Cao et al., 2023; Wu et al., 2024b; Cinquini et al., 2025).

- *Causal discovery.* Inferring causal directed acyclic graphs (DAGs) from observational time series has also been actively explored. Existing approaches include statistical dependence–based tests (Melkas et al., 2021; Bussmann et al., 2021), optimization-based formulations (Pamfil et al., 2020), and deep learning–based frameworks (Tank et al., 2021; Yin and Barucca, 2022; Zhong et al., 2023; Cheng et al.; Faruque et al., 2024; Cheng et al., 2024), which collectively advance the recovery of temporal causal structures under various assumptions.

However, existing causal effects on time-series methods focus mainly on discrete, fixed-time actions and estimate short-term expected outcome differences. Moreover, to our knowledge, no existing work has modeled the counterfactual trajectory, despite its importance for reliable decision making. Therefore, there remains a need for a generative model that explicitly embeds a causal DAG in time-series, supports interventions on individual continuous variables at arbitrary times, and yields coherent interventional and counterfactual predictions across the system.

## 2 TIME-CONDITIONED FLOW ON DAG

### 2.1 SETTINGS AND GOALS

We consider a $K$-dimensional multivariate time series evolving over a causal directed acyclic graph (DAG) with nodes $\{1, \ldots, K\}$ in a topologically sorted order. We denote by $X_{i,t} \in \mathbb{R}$ the value of node $i$ at time-series step $t$. Let $\mathbf{X}_t := \{X_{1,t}, \ldots, X_{K,t}\}$ denote the collection of all nodes at time $t$, and let $X_{\mathrm{pa}(i),t} := \{X_{j,t} : j \in \mathrm{pa}(i)\}$ denote the values of node $i$'s parents at time $t$.

The parent set $\mathrm{pa}(i)$ encodes the *structural dependencies* that remain fixed—for instance, in a physical system, upstream components serve as permanent causal parents of downstream ones. At each step $t$, the value $X_{i,t}$ depends on its past trajectory $X_{i,t-}$ and the past trajectories of its parents $X_{\mathrm{pa}(i),t-}$:

$$X_{i,t-} := \{X_{i,s}, s < t\}, \quad X_{\mathrm{pa}(i),t-} := \{X_{j,s}, j \in \mathrm{pa}(i), s < t\}. \tag{1}$$

We assume no within-time-step causal effects, meaning that all causal influences occur with at least one time-step lag. We formalize the dynamics by the structural causal model (SCM):

$$X_{i,t} := f_i(X_{i,t-}, X_{\mathrm{pa}(i),t-}, U_{i,t}), \tag{2}$$

where $U_{i,t}$ is an exogenous noise independent across nodes and time. Here, $f_i$ is a causal mechanism that takes as inputs the histories of $i$ and its parents, plus an exogenous noise.

For training and evaluation, each sequence $\{\mathbf{X}_1, \ldots, \mathbf{X}_T\}$ is divided into a context window $\{\mathbf{X}_1, \ldots, \mathbf{X}_\tau\}$ used for conditioning and a forecasting window $\{\mathbf{X}_{\tau+1}, \ldots, \mathbf{X}_T\}$ used for prediction. The context window consists of observed past values and is never intervened on. Interventions, if present, are applied within the forecasting window and influence all subsequent steps that we aim to predict. We use uppercase for random variables $\mathbf{X}_t$ and lowercase for realizations $\mathbf{x}_t$.

**Goals.** Our goal is to develop a framework for causal time-series prediction that answers:

- *Interventional forecasting.* We denote the intervention schedule by $\mathcal{I} \subseteq [K] \times \{\tau + 1, \ldots, T\}$, where each $(i, t) \in \mathcal{I}$ specifies an intervention on node $i$ at time $t$. For each $(i, t) \in \mathcal{I}$, the variable $X_{i,t}$ is intervened to the value $\gamma_{i,t}$, forming the collection $\gamma_{\mathcal{I}} := \{\gamma_{i,t}\}_{(i,t)\in\mathcal{I}}$. Each intervention replaces the system's natural evolution with a fixed value $\mathrm{do}(X_{i,t} := \gamma_{i,t})$, collectively written as $\mathrm{do}(X_{\mathcal{I}} := \gamma_{\mathcal{I}})$. This defines the interventional distribution:

$$p(\mathbf{X}_{\tau+1:T} \mid \mathbf{x}_{1:\tau}, \mathrm{do}(X_{\mathcal{I}} := \gamma_{\mathcal{I}})). \tag{3}$$

When $\mathcal{I} = \emptyset$, the model reduces to standard *observational forecasting*.

- *Counterfactual forecasting.* Given a factual trajectory $\mathbf{x}_{\tau+1:T}^{\mathrm{F}}$, we aim to answer: "What would this specific sequence have looked like if variables $X_{\mathcal{I}}$ had instead been set to $\gamma_{\mathcal{I}}$?" This induces the counterfactual distribution:

$$p(\mathbf{X}_{\tau+1:T}^{\mathrm{CF}} \mid \mathbf{x}_{1:\tau}, \mathbf{x}_{\tau+1:T}^{\mathrm{F}}, \mathrm{do}(X_{\mathcal{I}} := \gamma_{\mathcal{I}})). \tag{4}$$

## 2.2 Continuous normalizing flow foundations

We first introduce the continuous normalizing flow (CNF) foundation of DoFlow. It autoregressively generates future values, with each step conditioned on past temporal information. For each node $i$ in the causal DAG, we learn a *time-conditioned CNF* that models the conditional distribution of node $i$ at each time step $t$, given its own and its parents' past temporal context. This context is summarized by a hidden state $H_{i,t-1}$, which aggregates information from the historical sequences $X_{i,t-}$ and $X_{\mathrm{pa}(i),t-}$. In practice, $H_{i,t-1}$ is obtained using RNN-based encoders, as detailed in Section 2.3.

At each time-series step $t$, a CNF defines a continuous transformation between the target distribution (at $s{=}0$, corresponding to the target data $X_{i,t}$) and a base distribution (at $s{=}1$, typically $\mathcal{N}(0,1)$) through a neural ordinary differential equation (ODE) (Chen et al., 2018). Here, $s$ denotes the continuous ODE time, which is distinct from the time-series index $t$.

$$\frac{dx_{i,t}(s)}{ds} = v_i(x_{i,t}(s), s; H_{i,t-1}), \quad s \in [0,1], \tag{5}$$

where the velocity field $v_i$ is parameterized by a neural network. To predict $x_{i,t}$, we sample $z = x_{i,t}(1) \sim \mathcal{N}(0,1)$ and integrate the ODE from $s = 1$ to $s = 0$. We train the velocity field $v_i$ using Conditional Flow Matching (CFM) loss (Lipman et al., 2023), which directly regresses $v_i$ onto an analytically defined reference velocity field. Specifically, a reference path $\phi$ interpolates between a data sample $x_{i,t}$ and a base sample $z$ (typically using linear interpolation), whose derivative $\partial_s \phi$ defines the reference velocity. The model minimizes an $L_2$ loss between $v_i$ and $\partial_s \phi$, allowing efficient training while preserving the invertible mapping from base distribution to data distribution.

## 2.3 Time-conditioned continuous normalizing flow

**RNN to summarize past histories.** We learn a continuous normalizing flow (CNF) for each node $i$ to autoregressively predict $X_{i,t}$ for $t \in \{\tau + 1, \ldots, T\}$, conditioning each step on the node's past history and its parents' past histories. An RNN (can be either an LSTM or GRU (Dey and Salem, 2017)) is employed to summarize histories of node $i$ via a recurrent state $h_{i,t}$:

$$h_{i,t} = \mathrm{RNN}(x_{i,t}, h_{i,t-1}). \tag{6}$$

Therefore, for each node $i$ at time $t$, DoFlow is conditioned on the concatenated hidden states:

$$H_{i,t-1} := \mathrm{concat}(h_{i,t-1}, h_{\mathrm{pa}(i),t-1}), \tag{7}$$

where $h_{\mathrm{pa}(i),t-1} := \{h_{j,t-1} : j \in \mathrm{pa}(i)\}$.

At forecasting time, we use $\hat{H}_{i,t}$ (for observational or interventional forecasting) or $\hat{H}_{i,t}^{\mathrm{CF}}$ (for counterfactual forecasting) to denote the hidden states updated from the model predicted values. Importantly, since $\hat{H}_{i,t-1}$ (or $\hat{H}_{i,t-1}^{\mathrm{CF}}$) is autoregressively updated from the model's generated or intervened values, it carries information about any past interventions applied to node $i$ or its parents. Besides, we denote $H_{i,t}^{\mathrm{F}}$ as the factual hidden state computed from the observed factual trajectory.

**Time-conditioned CNF.** We train a separate CNF for each node $i$ that are shared across time-series steps $t$, conditioning each step on $H_{i,t-1}$. The Neural ODE of the conditional CNF is defined as:

$$\frac{dx_{i,t}(s)}{ds} = v_i(x_{i,t}(s), s; H_{i,t-1}), \tag{8}$$

$s \in [0,1], t \in \{\tau + 1, \ldots, T\}$, which connects samples from the time-series distribution at time $t$ (at $s = 0$) with the base distribution $\mathcal{N}(0,1)$ (at $s = 1$).

**Training loss.** At the start of the forecasting window, $H_{i,\tau}$ is initialized from the context sequence $(X_{i,1:\tau}, X_{\mathrm{pa}(i),1:\tau})$. For later steps $t \geq \tau + 1$, the hidden states $H_{i,t}$ are updated autoregressively during training using the observed values $(X_{i,t}, X_{\mathrm{pa}(i),t})$. The training loss is calculated over the entire forecasting window. For each $t \in \{\tau + 1, \ldots, T\}$, we define the reference path $\phi$ as a straight-line interpolation between the training sample $x_{i,t}$ and Gaussian base sample $z \sim \mathcal{N}(0, I)$:

$$\phi(x_{i,t}, z; s) := (1 - s)\, x_{i,t} + s\, z, \quad \partial_s \phi(x_{i,t}, z; s) = z - x_{i,t}, \tag{9}$$

with $s \sim \mathcal{U}[0,1]$.

Here, $\partial_s \phi$ represents the reference velocity field, which the model's learned velocity field $v_i$ aims to approximate. Training thus minimizes the squared $L_2$ distance between $v_i$ and $\partial_s \phi$. Therefore, the training loss of the flow using conditional flow matching (Lipman et al., 2023) becomes:

$$\mathcal{L}_{\text{CFM}}(\theta) = \mathbb{E}_{\mathbf{X}_{1:T} \sim p_{\mathcal{X}}} \left[ \frac{1}{K(T-\tau)} \sum_{i=1}^{K} \sum_{t=\tau+1}^{T} \right.$$

$$\left. \mathbb{E}_{s \sim \mathcal{U}[0,1],\, z \sim \mathcal{N}(0,I)} \big\| v_i\big(\phi(x_{i,t}, z; s),\, s;\, H_{i,t-1}\big) - \partial_s \phi(x_{i,t}, z; s)\big\|_2^2 \right], \tag{10}$$

where $\theta$ encompasses both the parameters of the velocity field $v_i$ and the RNN parameter.

After training, we obtain a velocity field $v_i$ for each node $i$ on the causal DAG. We next define the forward and reverse processes of the Neural ODE.

**Forward process.** We treat the forward process as an encoding operation, denoted by the function $Z_t := \Phi_\theta(X_t; H_{t-1})$. Given the velocity field $v_i$, the forward process pushes an observed factual outcome $x_{i,t}^{\text{F}}$ (at $s = 0$) to a latent embedding $z_{i,t}^{\text{F}}$ (at $s = 1$), conditioned on a factual hidden state $H_{i,t-1}^{\text{F}}$ that summarizes the past observed factual data. Formally,

$$z_{i,t}^{\text{F}} := \Phi_\theta(x_{i,t}^{\text{F}}; H_{i,t-1}^{\text{F}}) = x_{i,t}^{\text{F}} + \int_0^1 v_i\big(x_{i,t}(s),\, s;\, H_{i,t-1}^{\text{F}}\big)\, ds, \tag{11}$$

with $x_{i,t}(0) = x_{i,t}^{\text{F}}$. Through this process, $z_{i,t}^{\text{F}}$ encodes the information of the factual observation $x_{i,t}^{\text{F}}$.

**Reverse process.** We treat the reverse process as a decoding operation. It is initialized differently for interventional (Section 3.1) and counterfactual prediction (Section 3.2). In general, to predict the value of node $i$ at time $t$, given a latent representation $z_{i,t}$, which can be either sampled from $\mathcal{N}(0,1)$ or obtained by encoding a factual sample, the reverse process is defined as:

$$\hat{x}_{i,t} := \Phi_\theta^{-1}(z_{i,t}; \hat{H}_{i,t-1}) = z_{i,t} - \int_0^1 v_i\big(x_{i,t}(s),\, s;\, \hat{H}_{i,t-1}\big)\, ds, \tag{12}$$

with $x_{i,t}(1) = z_{i,t}$. Here, $\hat{H}_{i,t}$ is autoregressively updated using the predicted values $\hat{x}_{i,t}$ and $\hat{x}_{\text{pa}(i),t}$, and serves as the conditioning state for the next time step.

## 3 INTERVENTIONAL AND COUNTERFACTUAL PREDICTIONS

### 3.1 OBSERVATIONAL AND INTERVENTIONAL PREDICTION

At inference time, we forecast each node's value one step ahead using the reverse process conditioned on the latest hidden states. Assume that our intervention schedule is $\mathcal{I}$ with intervened values $\{\gamma_{i,t}\}$, and the purely observational case is given by $\mathcal{I} = \emptyset$.

The forecasting at time step $t$ is proceeded in a topologically sorted order, meaning that parent nodes are forecasted first, followed by their children. For a node $i$ intervened at time $t$ with value $\gamma_{i,t}$, the forecast is fixed to the intervened value, i.e., $\hat{x}_{i,t} \leftarrow \gamma_{i,t}$. For non-intervened nodes, the forecast $\hat{x}_{i,t}$ is generated by the flow model using reverse process (Eq. 12), conditioned on its hidden state $\hat{H}_{i,t-1}$, which is updated using the previous forecasts $\hat{x}_{i,t-1}$ and $\hat{x}_{\text{pa}(i),t-1}$. The overall procedure for observational and interventional forecasting over the time series is summarized in Algorithm 1. An illustrative figure is shown in Panel B of Figure 1.

---

**Algorithm 1:** Time Series Observational/Interventional Forecasting

1: **Input:** Context window $\{x_{i,1:\tau}\}_{i=1}^K$; intervention schedule $\mathcal{I}$ with values $\{\gamma_{i,t}\}$
2: Initialize hidden states $\hat{H}_{i,\tau}$ with $(x_{i,1:\tau}, x_{\text{pa}(i),1:\tau})$ for all $i = 1, \ldots, K$
3: **for** $t = \tau + 1$ **to** $T$ **do**
4:      **for** $i = 1, \ldots, K$ **do** {topological order}
5:          **if** $(i,t) \in \mathcal{I}$ **then**
6:              $\hat{x}_{i,t} \leftarrow \gamma_{i,t}$
7:          **else**
8:              Sample $z_{i,t} \sim \mathcal{N}(0,1)$
9:              $\hat{x}_{i,t} \leftarrow \Phi_\theta^{-1}(z_{i,t}; \hat{H}_{i,t-1})$ {Eq. (12)}
10:          **end if**
11:          $h_{i,t} \leftarrow \text{RNN}(\hat{x}_{i,t}, h_{i,t-1})$
12:          $\hat{H}_{i,t} \leftarrow (h_{i,t}, h_{\text{pa}(i),t})$
13:      **end for**
14: **end for**
15: **Output:** $\{\hat{x}_{i,t}\}_{i=1..K,\, t=\tau+1,..,T}$

---

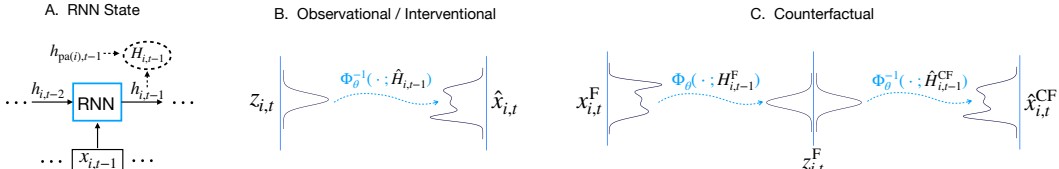

Figure 1: **(A)** RNN State Update. **(B)** Observational/Interventional Forecasting. Forecasts are generated by decoding from latent $z_{i,t} \sim N(0,1)$, conditioned on $\hat{H}_{i,t-1}$ updated with the last predicted $(\hat{x}_{i,t-1}, \hat{x}_{\mathrm{pa}(i),t-1})$. **(C)** Counterfactual Forecasting. A factual observation $x_{i,t}^{\mathrm{F}}$ is encoded with its factual state $H_{i,t-1}^{\mathrm{F}}$ into $z_{i,t}^{\mathrm{F}}$, then decoded under the counterfactual state $\hat{H}_{i,t-1}^{\mathrm{CF}}$ to yield $\hat{x}_{i,t}^{\mathrm{CF}}$. Factual states $H_{i,t-1}^{\mathrm{F}}$ are updated from observed factual $(x_{i,t-1}^{\mathrm{F}}, x_{\mathrm{pa}(i),t-1}^{\mathrm{F}})$, while counterfactual states $\hat{H}_{i,t-1}^{\mathrm{CF}}$ are updated from the previously predicted $(\hat{x}_{i,t-1}^{\mathrm{CF}}, \hat{x}_{\mathrm{pa}(i),t-1}^{\mathrm{CF}})$.

### 3.2 Time Series Forecasting for Counterfactual Queries

Counterfactual forecasting follows the standard abduction–action–prediction procedure. Given a factual trajectory $\{x_{i,\tau+1:T}^{\mathrm{F}}\}_{i=1}^{K}$, we proceed as follows: (i) *abduction* – we infer latent variables by encoding each observed factual value into its latent representation $z_{i,t}^{\mathrm{F}}$ through the forward process (11), conditioned on a factual hidden state $H_{i,t-1}^{\mathrm{F}}$ that summarizes past factual observations; (ii) *action* – we apply the specified intervention schedule $\mathcal{I}$; and (iii) *prediction* – we generate the counterfactual trajectory $\hat{x}_{i,t}^{\mathrm{CF}}$ through the reverse process (12), starting from the abducted latent representations $z_{i,t}^{\mathrm{F}}$.

Specifically, we first compute the factual hidden states $\{H_{i,t}^{\mathrm{F}}\}_{t=\tau}^{T-1}$ for each $i$ from the observed context $\{x_{i,1:\tau}\}_{i=1}^{K}$ and the observed factual trajectory $\{x_{i,\tau+1:T}^{\mathrm{F}}\}_{i=1}^{K}$. These factual states are used only for encoding factual values into their latent representations, which serve as the starting point for counterfactual decoding. At each time-series step $t$, nodes are predicted in topological order. For $(i,t) \in \mathcal{I}$, the counterfactual forecast is set as $\hat{x}_{i,t}^{\mathrm{CF}} := \gamma_{i,t}$, and its RNN state $h_{i,t}$ is updated accordingly. For non-intervened nodes, the factual value $x_{i,t}^{\mathrm{F}}$ is first encoded as $z_{i,t}^{\mathrm{F}} = \Phi_\theta(x_{i,t}^{\mathrm{F}}; H_{i,t-1}^{\mathrm{F}})$, then decoded under the counterfactual hidden state as $\hat{x}_{i,t}^{\mathrm{CF}} = \Phi_\theta^{-1}(z_{i,t}^{\mathrm{F}}; \hat{H}_{i,t-1}^{\mathrm{CF}})$, followed by updating $\hat{H}_{i,t}^{\mathrm{CF}}$. The overall counterfactual generation procedure is summarized in Algorithm 2 and illustrated in Panel C of Figure 1.

---

**Algorithm 2:** Counterfactual Time Series Generation

1: **Input:** Context window $\{x_{i,1:\tau}\}_{i=1}^{K}$; factual sample $\{x_{i,\tau+1:T}^{\mathrm{F}}\}_{i=1}^{K}$; intervention schedule $\mathcal{I}$ with values $\{\gamma_{i,t}\}$
2: Obtain factual hidden states $\{H_{i,t}^{\mathrm{F}}\}_{t=\tau}^{T-1}$ for each $i$ from context $\{x_{i,1:\tau}\}_{i=1}^{K}$ and observed factual $\{x_{i,\tau+1:T}^{\mathrm{F}}\}_{i=1}^{K}$
3: Initialize counterfactual hidden states $\hat{H}_{i,\tau}^{\mathrm{CF}}$ with context $(x_{i,1:\tau}, x_{\mathrm{pa}(i),1:\tau})$ for all $i = 1, \ldots, K$
4: **for** $t = \tau + 1$ **to** $T$ **do**
5:   **for** $i = 1, \ldots, K$ **do** {topological order}
6:     **if** $(i,t) \in \mathcal{I}$ **then**
7:       $\hat{x}_{i,t}^{\mathrm{CF}} \leftarrow \gamma_{i,t}$
8:     **else**
9:       $z_{i,t}^{\mathrm{F}} \leftarrow \Phi_\theta(x_{i,t}^{\mathrm{F}}, H_{i,t-1}^{\mathrm{F}})$    {Eq. (11); Abduction}
10:      $\hat{x}_{i,t}^{\mathrm{CF}} \leftarrow \Phi_\theta^{-1}(z_{i,t}^{\mathrm{F}}, \hat{H}_{i,t-1}^{\mathrm{CF}})$    {Eq. (12); Action-Prediction}
11:     **end if**
12:     $h_{i,t} \leftarrow \mathrm{RNN}(\hat{x}_{i,t}^{\mathrm{CF}}, h_{i,t-1})$
13:     $\hat{H}_{i,t}^{\mathrm{CF}} \leftarrow (h_{i,t}, h_{\mathrm{pa}(i),t})$
14:   **end for**
15: **end for**
16: **Output:** $\{\hat{x}_{i,t}^{\mathrm{CF}}\}_{i=1..K,\, t=\tau+1,..T}$

---

### 3.3 ADDITIONAL PROPERTY: LIKELIHOOD-BASED ANOMALY DETECTION

Another advantage of our framework is that it assigns an explicit log-density to its predicted future trajectory over the forecasting window. For a single node (index $i$ omitted for clarity), we denote this conditional density by $p_{\theta, X_{\tau+1:T}}(\cdot \mid \hat{H}_\tau)$, where $\hat{H}_\tau$ is the context state. In DoFlow, $z_{\tau+1:T}$ are base samples used to generate the forecast $\hat{x}_{\tau+1:T}$. The explicit form of this learned density, mapping from latent $z_{\tau+1:T}$ to predicted $\hat{x}_{\tau+1:T}$, is given as:

**Proposition 3.1.** *Given base samples $z_{\tau+1:T} \sim q(\cdot)$, the log-density of the generated time series obtained via the continuous normalizing flow is:*

$$\log p_{\theta, X_{\tau+1:T}}\left(\hat{x}_{\tau+1:T} \mid \hat{H}_\tau\right) = \sum_{t=\tau+1}^{T} \left[ \log q(z_t) + \int_0^1 \nabla \cdot v_\theta\left(x_t(s), s; \hat{H}_{t-1}\right) ds \right], \tag{13}$$

*where $\hat{x}_t = \Phi_\theta^{-1}(z_t; \hat{H}_{t-1})$, $t \in \{\tau+1, \dots, T\}$.*

**Anomaly Detection.**  Since anomalies deviate substantially from normal patterns, we expect the model to assign lower density to its *generated forecast trajectories* when conditioned on anomalous contexts.

## 4 THEORETICAL PROPERTIES

Our analysis assumes that the underlying SCM on the given DAG satisfies the conditions in Assumption B.1 (see Appendix B.1), which are standard and fundamental in causal inference (Pearl, 2009; Hernán and Robins, 2010; Imbens and Rubin, 2015; Javaloy et al., 2023; Chao et al., 2024). These assumptions ensure that the interventional and counterfactual distributions we study are well-defined.

In this section, we present a theoretical result on the counterfactual recovery of our algorithm under certain assumptions. We fix a node and omit the index $i$ for simplicity. Recall that the encoding function is defined as $\Phi_\theta : \mathcal{X} \times \mathcal{H} \to \mathcal{Z}$ and the decoding function as $\Phi_\theta^{-1} : \mathcal{Z} \times \mathcal{H} \to \mathcal{X}$, corresponding to (11) and (12), respectively. Also, the underlying structural causal model is given by $X_t := f(X_{t-}, X_{\mathrm{pa},t-}, U_t)$. In the following, we present a supporting result on the counterfactual recovery properties of DoFlow. We begin by introducing the following assumptions.

**Assumption 4.1.**
(A1) $U_t \perp\!\!\!\perp (X_{t-}, X_{\mathrm{pa},t-})$.
(A2) The structural causal equation $f(\cdot, U_t)$ is strictly monotone and continuous in $U_t$.
(A3) For the encoded latent variable $Z_t = \Phi_\theta(X_t; H_{t-1})$, the conditional distribution satisfies $p_\theta(Z_t \mid H_{t-1}) = q(Z_t)$.

**Remark 4.2.** Each node $X_t \in \mathbb{R}$, so (A2) is assumed in the univariate case. It is automatically satisfied under additive SCMs, i.e., $X_t = f^*(X_{t-}, X_{\mathrm{pa},t-}) + U_t$. Under certain identifiability conditions, it also holds for non-linear models (Zhang and Hyvarinen, 2012; Strobl and Lasko, 2023). Under (A2), for any fixed $(X_{t-}, X_{\mathrm{pa},t-})$, the map $u \mapsto f(X_{t-}, X_{\mathrm{pa},t-}, u)$ is a bijection on the support of $U_t$. One may notice that (A1)–(A2) mirror those in Bijective Generation Mechanisms (BGM) (Nasr-Esfahany et al., 2023), which establish model-agnostic identifiability with an additional assumption on distribution matching. In contrast, our Corollary 4.5 provides model-specific, pointwise recovery for our CNF under (A1)–(A3) without requiring distribution matching; see Appendix B.2.

Note that (A3) implies that the encoded $Z_t = \Phi_\theta(X_t; H_{t-1})$ is statistically independent of $H_{t-1}$, and equivalently of $(X_{t-}, X_{\mathrm{pa},t-})$, in distribution. In the infinite-data limit with exact training, the continuous normalizing flow maps every $X_t$, conditioned on any fixed $(X_{t-}, X_{\mathrm{pa},t-})$, to the same base distribution $q(Z_t) = N(0,1)$, so that (A3) holds exactly.

Under Assumption 4.1, we present the first result in this paper:

**Proposition 4.3** (Encoded as a function of the exogenous noise $U_t$)**.** *Let Assumption 4.1 hold. Without loss of generality, suppose the exogenous noise $U_t \sim \mathrm{Unif}[0,1]$. At each time $t$, the observed variable is generated by the structural causal model $X_t = f(X_{t-}, X_{\mathrm{pa},t-}, U_t)$, and that the flow encoder produces $Z_t = \Phi_\theta(X_t; H_{t-1})$. Then there exists a continuously differentiable bijection $g : \mathcal{U} \to \mathcal{Z}$, functionally invariant to $H_{t-1}$, such that,*

$$Z_t = \Phi_\theta\left(X_t; H_{t-1}\right) = \Phi_\theta\left(f(X_{t-}, X_{\mathrm{pa},t-}, U_t); H_{t-1}\right) = g\left(U_t\right) \quad a.s. \tag{14}$$

**Remark 4.4.** Proposition 4.3 states that $Z_t$ is a function of the exogenous noise $U_t$, and for convenience assumes $U_t \sim \mathrm{Unif}[0,1]$. This distributional assumption can be relaxed to other noise distributions. For example, if $Z \sim \mathcal{N}(0,1)$ with CDF $F$, then $U = F(Z) \sim \mathrm{Unif}[0,1]$, and any assignment $f(\cdot, U)$ can equivalently be written as $\tilde{f}(\cdot, Z) = f(\cdot, F(Z))$.

Following Proposition 4.3, we state a counterfactual recovery result under monotone SCMs. Given an intervention schedule $\mathcal{I}$ with values $\{\gamma_t\}$, the true counterfactual process is defined recursively as:

$$X_t^{\mathrm{CF}} = \begin{cases} \gamma_t, & t \in \mathcal{I}, \\ f(X_{t-}^{\mathrm{CF}}, X_{\mathrm{pa},t-}^{\mathrm{CF}}, U_t), & \text{otherwise,} \end{cases} \tag{15}$$

where $U_t$ is *abducted* from the factual sample $X_t^{\mathrm{F}}$. Also, $\hat{X}_{t-1}^{\mathrm{CF}}$ and $\hat{X}_{\mathrm{pa},t-1}^{\mathrm{CF}}$ denote the model predicted counterfactual values by Algorithm 2. We can now state the following result:

**Corollary 4.5** (Counterfactual recovery). *Let Assumption 4.1 hold. Consider a factual sample generated by the structural causal model $X_t^{\mathrm{F}} = f(X_{t-}, X_{\mathrm{pa},t-}, U_t)$, and let its encoded latent be $Z_t^{\mathrm{F}} := \Phi_\theta\big(X_t^{\mathrm{F}}; H_{t-1}^{\mathrm{F}}\big)$. At time step $t$, we obtain the counterfactual hidden state $\hat{H}_{t-1}^{\mathrm{CF}}$ from the predicted counterfactual past $(\hat{X}_{t-}^{\mathrm{CF}}, \hat{X}_{\mathrm{pa},t-}^{\mathrm{CF}})$. The decoder recovers the true counterfactual at time step $t$ almost surely:*

$$\hat{X}_t^{\mathrm{CF}} := \Phi_\theta^{-1}\big(Z_t^{\mathrm{F}}; \hat{H}_{t-1}^{\mathrm{CF}}\big) = X_t^{\mathrm{CF}}. \tag{16}$$

## 5 EXPERIMENTS

In this section, we evaluate DoFlow on observational forecasting, interventional and counterfactual forecasting, and anomaly detection, on both synthetic and real-world datasets. Our code is publicly available at `https://github.com/StatFusion/DoFlow_Causal_Time_Series`.

In real-world, the ground-truth counterfactual is never observable, and the ground-truth interventional is only observable if interventions are actively conducted under a correct causal DAG. Therefore, we rely on synthetic experiments to obtain quantitative performance metrics. For real-world evaluation, we assess our model on interventional queries and anomaly detection using hydropower datasets from Argonne National Laboratory, and on interventional treatment effect estimation using the cancer-treatment dataset from Bica et al. (2020a).

### 5.1 SYNTHETIC DATA EXPERIMENTS

We consider both the additive noise model, i.e., $f_i(X_{i,t-}, X_{\mathrm{pa}(i),t-}, U_{i,t}) = f_i^*(X_{i,t-}, X_{\mathrm{pa}(i),t-}) + U_{i,t}$, which satisfies Assumption 4.1 and supports the counterfactual recovery result, as well as more general non-linear and non-additive (NLNA) cases of $f_i(X_{i,t-}, X_{\mathrm{pa}(i),t-}, U_{i,t})$ to test the model's robustness beyond the scope of our counterfactual recovery result.

We evaluate the model across multiple structurally diverse causal DAGs using Root Mean Squared Error (RMSE), Maximum Mean Discrepancy (MMD), and Continuous Ranked Probability Score (CRPS). RMSE is reported for all settings. MMD and CRPS are computed only for observational and interventional forecasting, since counterfactuals in DoFlow yield a single deterministic trajectory. Detailed simulation setups and metric definitions are provided in Appendix D.

For baseline comparisons in observational forecasting, we consider: a pure RNN-based method, the Gated Recurrent Unit (GRU) (Che et al., 2018); transformer-based methods, including the Temporal Fusion Transformer (TFT) (Lim et al., 2021) and the Time Series Dense Encoder (TiDE) (Das et al., 2024); and an all-MLP method with a specialized contextual mixing structure, the Time Series Mixer (TSMixer) (Ekambaram et al., 2023). We also compare against probabilistic methods, including DeepVAR (Salinas et al., 2019) — a deep RNN-based model with multivariate Gaussian outputs, and MQF2 (Kan et al., 2022) — a convex deep neural network that learns multivariate quantile functions. We extend our acknowledgements to the Python packages Darts (Herzen et al., 2022) and GluonTS (Alexandrov et al., 2020), which we use to directly test several modern baselines in this paper.

To our knowledge, comparatively few works tackle interventional or counterfactual time-series forecasting on a causal DAG. Nevertheless, we can adapt strong observational forecasters for interventional task by training a separate model for each node in the causal DAG, conditioned on its parents; see Appendix E.1. However, counterfactual generation remains challenging for these baselines.

Table 1: RMSE for observational, interventional, and counterfactual time series forecasting across causal structures: Tree, Diamond, and FC-Layer. Results for the Chain structure are provided in Table 6 (Appendix). MMD and CRPS results are provided in Table 4 and Table 5 (Appendix), respectively. Reported values are averaged over 50 test batches, each containing 128 test series.

| | Tree | | | | | | Diamond | | | | | | FC-Layer | | | | | |
|---|---|---|---|---|---|---|---|---|---|---|---|---|---|---|---|---|---|---|
| | Additive | | | NLNA | | | Additive | | | NLNA | | | Additive | | | NLNA | | |
| | Obs. | Int. | CF. | Obs. | Int. | CF. | Obs. | Int. | CF. | Obs. | Int. | CF. | Obs. | Int. | CF. | Obs. | Int. | CF. |
| **DoFlow** | $0.57_{\pm.09}$ | $0.54_{\pm.10}$ | $0.11_{\pm.02}$ | $0.59_{\pm.13}$ | $0.62_{\pm.12}$ | $0.14_{\pm.02}$ | $0.55_{\pm.10}$ | $0.57_{\pm.08}$ | $0.12_{\pm.01}$ | $0.31_{\pm.07}$ | $0.38_{\pm.08}$ | $0.19_{\pm.03}$ | $0.39_{\pm.12}$ | $0.41_{\pm.13}$ | $0.16_{\pm.03}$ | $0.49_{\pm.08}$ | $0.36_{\pm.06}$ | $0.18_{\pm.04}$ |
| GRU | $0.65_{\pm.08}$ | $1.01_{\pm.10}$ | NA | $0.63_{\pm.07}$ | $1.04_{\pm.11}$ | NA | $0.58_{\pm.06}$ | $0.94_{\pm.11}$ | NA | $0.37_{\pm.05}$ | $0.99_{\pm.12}$ | NA | $\mathbf{0.38_{\pm.05}}$ | $0.72_{\pm.10}$ | NA | $0.58_{\pm.07}$ | $1.05_{\pm.13}$ | NA |
| TFT | $\mathbf{0.58_{\pm.11}}$ | $0.97_{\pm.17}$ | NA | $0.63_{\pm.07}$ | $1.01_{\pm.18}$ | NA | $0.63_{\pm.17}$ | $1.18_{\pm.21}$ | NA | $0.40_{\pm.08}$ | $1.09_{\pm.20}$ | NA | $0.47_{\pm.14}$ | $0.83_{\pm.16}$ | NA | $0.62_{\pm.15}$ | $1.02_{\pm.23}$ | NA |
| TiDE | $0.60_{\pm.13}$ | $1.15_{\pm.21}$ | NA | $0.68_{\pm.14}$ | $1.13_{\pm.20}$ | NA | $\mathbf{0.50_{\pm.12}}$ | $1.05_{\pm.19}$ | NA | $0.43_{\pm.12}$ | $0.99_{\pm.16}$ | NA | $0.43_{\pm.12}$ | $0.75_{\pm.14}$ | NA | $0.66_{\pm.17}$ | $1.10_{\pm.20}$ | NA |
| TSMixer | $0.63_{\pm.13}$ | $1.08_{\pm.18}$ | NA | $0.65_{\pm.13}$ | $1.07_{\pm.18}$ | NA | $\mathbf{0.49_{\pm.10}}$ | $1.12_{\pm.20}$ | NA | $0.35_{\pm.11}$ | $0.97_{\pm.15}$ | NA | $0.42_{\pm.11}$ | $0.79_{\pm.15}$ | NA | $0.61_{\pm.15}$ | $1.13_{\pm.19}$ | NA |
| DeepVAR | $0.64_{\pm.07}$ | $0.74_{\pm.12}$ | NA | $0.65_{\pm.09}$ | $0.86_{\pm.15}$ | NA | $0.68_{\pm.09}$ | $0.86_{\pm.17}$ | NA | $0.45_{\pm.08}$ | $0.94_{\pm.16}$ | NA | $0.54_{\pm.10}$ | $1.17_{\pm.18}$ | NA | $0.69_{\pm.11}$ | $1.57_{\pm.21}$ | NA |
| MQF2 | $\mathbf{0.58_{\pm.10}}$ | $1.23_{\pm.19}$ | NA | $0.67_{\pm.11}$ | $1.30_{\pm.21}$ | NA | $0.64_{\pm.12}$ | $1.20_{\pm.16}$ | NA | $0.38_{\pm.06}$ | $1.17_{\pm.18}$ | NA | $0.50_{\pm.09}$ | $1.09_{\pm.12}$ | NA | $0.57_{\pm.10}$ | $1.33_{\pm.22}$ | NA |

Tables 1 (main) and 4, 5 (Appendix) report the RMSE, MMD, and CRPS results for observational, interventional, and counterfactual time-series forecasting across multiple causal structures (see Appendix D). DoFlow consistently delivers strong performance in standard observational forecasting and interventional forecasting compared with our adapted baselines, and uniquely supports counterfactual forecasting. We present visual results for both interventional and counterfactual forecasting in Figure 2 (main text), and in Figures 8 and 9 (Appendix).

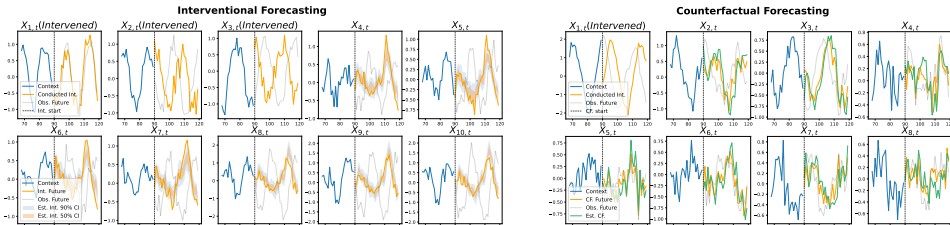

Figure 2: *Left:* "**Layer**" interventional forecasting results. Nodes $X_{1,t}$, $X_{2,t}$, and $X_{3,t}$ are intervened. DoFlow provides 50% and 90% prediction intervals; the orange lines indicate the true interventional future. *Right:* "**Tree**" counterfactual forecasting results. Node $X_{1,t}$ is intervened. DoFlow provides a single forecast in green; the orange lines indicate the true counterfactual future.

## 5.2 REAL APPLICATION: HYDROPOWER SYSTEM

We evaluate DoFlow on real-world hydropower time-series data from Argonne National Laboratory. In this system, water drives a turbine that powers a generator and passes through a transformer before reaching the grid, with control systems monitoring the process. Signals such as water flow, vibration, and electric current are recorded from each component, forming a natural DAG as shown in Fig. 10.

Our evaluation focuses on two tasks: (1) whether DoFlow can accurately forecast each component's time series under interventional queries; and (2) whether DoFlow can accurately detect power outages in advance using log-density, as discussed in Section 3.3.

Figure 3 illustrates interventional forecasting in the hydropower system during a *true power outage*. At this point, the turbine signals ($X_{1,t}$, $X_{2,t}$) break down, leading the entire system into a forced outage. Notably, DoFlow successfully predicts the characteristic "spikes" in the generator signals ($X_{4,t}$ and $X_{5,t}$) that follow the turbine failure, demonstrating that our DAG-based approach captures the complex causal relationships across system. Quantitative metrics, together with the anomaly detection visualizations from the second task, are provided in the Appendix F.3.

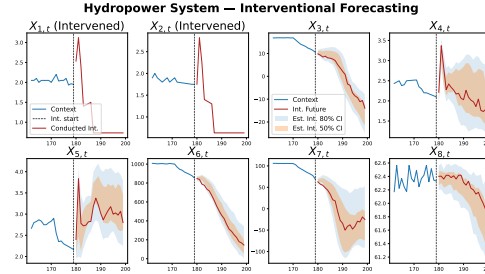

Figure 3: Hydropower – Interventional.

### 5.3 REAL APPLICATION: CANCER TREATMENT OUTCOMES

Our DoFlow can naturally model interventional forecasting for causal treatment effect estimation in time series. The *causal effects on time series* were thoroughly introduced in Section 1.1. We evaluate DoFlow on the cancer-treatment benchmark of Bica et al. (2020a), which contains daily patient-level tumor volumes (outcome) and administered therapies (actions). The action variables $\mathbf{X}_{t-1} = \{X_{i,t-1}\}_{i=1}^4$ (chemotherapy assignment, radiotherapy assignment, chemotherapy dosage, and radiotherapy dosage) act as causal parents of the outcome $Y_t$ (tumor volume) on the causal DAG. For each test patient, we use the first $55$ days as observational context and perform interventional rollouts for the next 5 days under ten distinct treatment options, yielding predicted outcomes $\hat{Y}_{t+1:t+5}$. The normalized RMSE between the true $Y_{t+1:t+5}$ and predicted $\hat{Y}_{t+1:t+5}$ is reported in Appendix F.4.

The detailed settings and results are provided in Appendix F.4. Notably, DoFlow achieves substantial improvements over prior baselines on this causal treatment effect task, as summarized in Table 8.

### 5.4 DISCUSSIONS AND LIMITATIONS

- *Training Efficiency.* DoFlow trains a separate flow per node in the causal DAG, but each network can be much shallower than transformer models. As shown in Tables 2 and 3, its total model size remains modest, and its training and sampling times are comparable to modern baselines.
- *Interventions.* We adapt modern baselines for conditional intervention generation, but DoFlow performs notably better: its RNN–flow design jointly encodes causal histories and propagates interventions through recurrent states, while the flow backbone enables coherent decoding.
- *Counterfactuals.* As discussed in the Introduction, DoFlow is among the early approaches for generating counterfactual time-series on a causal DAG, complementing work on treatment effects and causal discovery on time series. Such counterfactual generation is crucial for modeling component relationships and enabling post-hoc analyses in crucial areas.
- *Limitations - Known DAG.* DoFlow currently assumes a known and correctly specified causal DAG, as well as causal sufficiency (see Assumption B.1). In practice, the DAG may come from known physical relationships or from upstream causal discovery methods as discussed in Section 1.1. A natural direction for future work is to couple DoFlow with time-series causal discovery or deconfounding causal generative models (Almodóvar et al., 2025; Xia et al., 2022), allowing the model to operate on learned temporal causal structures for more reliable causal reasoning.

## 6 CONCLUSIONS

We introduced DoFlow, a flow-based generative framework for causal time-series forecasting that unifies observational, interventional, and counterfactual queries on DAG-structured causal systems. Moreover, we provide a supporting counterfactual recovery result for DoFlow (Corollary 4.5) under certain assumptions. Experiments on various synthetic DAGs and real applications show strong observational forecasts and effective prediction under causal queries. This framework lays the foundation for domains such as energy, healthcare, and other areas where counterfactual and interventional forecasting is critical. The integration of causal structure and generative modeling represents a step toward a general theory of inference and control in complex dynamical environments.

### ACKNOWLEDGMENT

D.W. and Y.X. are partially supported by NSF CAREER CCF-1650913, NSF DMS-2134037, CMMI-2015787, CMMI-2112533, DMS-1938106, DMS-1830210, and the Coca-Cola Foundation. F.Q. is supported by the U.S. Department of Energy Advanced Grid Modeling Program under Grant DE-OE0000875.

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

## A    PROOFS

**Proposition 3.1** *Given base samples $z_{\tau+1:T} \sim q(\cdot)$, the log-density of the generated time series obtained via the continuous normalizing flow is:*

$$\log p_{\theta, X_{\tau+1:T}}\left(\hat{x}_{\tau+1:T} \mid \hat{H}_\tau\right) = \sum_{t=\tau+1}^{T} \left[\log q(z_t) + \int_0^1 \nabla \cdot v_\theta\left(x_t(s), s; \hat{H}_{t-1}\right) ds\right],$$

*where $\hat{x}_t = \Phi_\theta^{-1}(z_t; \hat{H}_{t-1})$, $t \in \{\tau+1, \ldots, T\}$.*

*Proof.* Samples from Continuous Normalizing Flows (CNFs) evolve according to the following Neural ODE:

$$\frac{dx_t(s)}{ds} = v(x_t(s), s; \hat{H}_{t-1}), \quad s \in [0, 1], \tag{17}$$

which induces a corresponding evolution of the sample density governed by Liouville's continuity equation (Chen et al., 2018):

$$\partial p_\theta(x_t, s \mid \hat{H}_{t-1}) + \nabla \cdot \big(p_\theta(x_t, s \mid \hat{H}_{t-1}) v(x_t, s; \hat{H}_{t-1})\big) = 0. \tag{18}$$

Here, $p_\theta(x_t(s), s)$ denotes the probability density of the sample $x_t(s)$ at ODE-time $s$.

Next, we have that the dynamics of the density $p_\theta(\cdot)$ governed by the velocity field $v_\theta(x_t, s; \hat{H}_{t-1})$ is given by:

$$\frac{d}{ds} \log p_\theta(x_t(s), s \mid \hat{H}_{t-1}) = \left. \frac{\nabla p_\theta \cdot \partial_s x_t(s) + \partial_s p_\theta}{p_\theta} \right|_{(x_t(s), s)} \tag{19}$$

$$= \left. \frac{\nabla p_\theta \cdot v - \nabla \cdot (p_\theta v)}{p_\theta} \right|_{(x_t(s), s)} \quad \text{(by (17) and (18))} \tag{20}$$

$$= \left. \frac{\nabla p_\theta \cdot v - (\nabla p_\theta \cdot v + p_\theta \nabla \cdot v)}{p_\theta} \right|_{(x_t(s), s)} \tag{21}$$

$$= -\nabla \cdot v. \tag{22}$$

Starting from a base sample $z_t \sim q(\cdot)$ and integrating from $s = 1$ to $s = 0$, we have:

$$\log p_{\theta, X_t}\big(\hat{x}_t \mid \hat{H}_{t-1}\big) = \log q(z_t) + \int_0^1 \nabla_x \cdot v\big(x_t(s), s; \hat{H}_{t-1}\big) \, ds. \tag{23}$$

By summing up all the log-densities within the forecasting window, we obtain:

$$\log p_{\theta, X_{\tau+1:T}}\big(\hat{x}_{\tau+1:T} \mid \hat{H}_\tau\big) = \sum_{t=\tau+1}^{T} \left[ \log q(z_t) + \int_0^1 \nabla \cdot v_\theta\big(x_t(s), s; \hat{H}_{t-1}\big) \, ds \right], \tag{24}$$

where $\hat{x}_t = \Phi_\theta^{-1}(z_t; \hat{H}_{t-1})$, $t \in \{\tau + 1, \ldots, T\}$. $\qquad \square$

**Proposition 4.3** (Encoded as a function of the exogenous noise $U$). *Let Assumption 4.1 hold. Without loss of generality, suppose the exogenous noise $U_t \sim \mathrm{Unif}[0, 1]$. At each time $t$, the observed variable is generated by the structural causal model $X_t = f(X_{t-}, X_{\mathrm{pa}, t-}, U_t)$, and that the flow encoder produces $Z_t = \Phi_\theta(X_t; H_{t-1})$. Then there exists a continuously differentiable bijection $g : \mathcal{U} \to \mathcal{Z}$, functionally invariant to $H_{t-1}$, such that,*

$$Z_t = \Phi_\theta\big(X_t; H_{t-1}\big) = \Phi_\theta\big(f(X_{t-}, X_{\mathrm{pa}, t-}, U_t); H_{t-1}\big) = g\big(U_t\big) \quad \textit{a.s.}$$

*Proof.* Fix a node $i$ (index suppressed) and a time $t$. We write the extended parental state as $S_t := (X_{t-}, X_{\mathrm{pa}, t-})$. Since the hidden state depends solely on $S_t$, we define

$$q_{S_t}(U_t) := Z_t = \Phi_\theta\big(f(X_{t-}, X_{\mathrm{pa}, t-}, U_t); H_{t-1}\big).$$

Therefore, our goal becomes proving that $Z_t = q_{S_t}(U_t)$ is a function invariant of $S_t$. By (A3) in Assumption 4.1, we have $Z_t = q_{S_t}(U_t) \perp\!\!\!\perp S_t$, and thus:

$$p_{Z_t \mid S_t = s_t}(z_t) = p_{Z_t}(z_t). \tag{25}$$

Because continuous normalizing flows are invertible, the encoding function $\Phi_\theta : \mathcal{X} \times \mathcal{H} \to \mathcal{Z}$ is invertible with respect to $\mathcal{X}$. In 1-D, this implies monotonicity of $\Phi_\theta$ in $X_t = f(X_{t-}, X_{\mathrm{pa}, t-}, U_t)$. Without loss of generality, we assume that $\Phi_\theta$ is strictly increasing in $X_t = f$. Moreover, by (A2),

since $f(\cdot, U_t)$ is strictly increasing in $U_t$, it follows by the composition rule that $q_{S_t}(U_t)$ is strictly increasing in $U_t$ and hence bijective in $[0, 1]$.

Since $Z_t = q_{S_t}(U_t)$, we may apply change of variables formula:

$$p_{Z_t | S_t = s_t}(z_t) \;=\; p_{U_t}\big(q_{s_t}^{-1}(z_t)\big)\left|\frac{d}{dz_t}q_{s_t}^{-1}(z_t)\right| = 1 \cdot \frac{d}{dz_t}q_{s_t}^{-1}(z_t), \tag{26}$$

where the last equation follows from the uniform distribution of $U_t$ and the fact that $p_{Z_t} > 0$. The absolute value is dropped because of the (WLOG) assumption that $q_{s_t}$ is strictly increasing.

Then because of (25), we have that for a fixed $z_t$, $\frac{d}{dz_t}q_{s_t}^{-1}(z_t)$ is the same for any pair $s_t = (x_{t-}, x_{\text{pa},t-})$. It follows that:

$$q_{s_t}^{-1}(z_t) = \int^{z_t} \frac{d}{dx}q_{s_t}^{-1}(x)dx = \int^{z_t} c(x)dx + c_{s_t}, \tag{27}$$

where $c(z_t) = \frac{d}{dz_t}q_{s_t}^{-1}(z_t)$ is independent of $s_t$, and $c_{s_t}$ is a constant for each $s_t$.

By re-inverting (27), we have:

$$q_{s_t}(u_t) = (q_{s_t}^{-1})^{-1}(u_t) = \underbrace{\left(\int^{\bullet} c(x)\,dx + c_{s_t}\right)^{-1}}_{:=G(\bullet)}(u_t) = G^{-1}\big(u_t - c_{s_t}\big). \tag{28}$$

Since $q_{S_t}$ is a bijection $[0, 1] \to \text{supp}(Z_t) = \{z \in \mathbb{R} : p_{Z_t}(z_t) > 0\}$, we have:

$$q_{s_t}(0) = \inf \ \text{supp}(Z_t), \quad q_{s_t}(1) = \sup \ \text{supp}(Z_t), \ \text{for any state } s_t. \tag{29}$$

Therefore,

$$q_{s_t}(0) = G^{-1}(-c_{s_t}) = \inf \ \text{supp}(Z_t). \tag{30}$$

The support of $Z_t$ does not depend on $S_t$ because of (25). As a result, for any state $s_t = (x_{t-}, x_{\text{pa},t-})$, we have that $c_{s_t} := c$ is a constant that does not depend on $s_t$.

Therefore, we can write

$$g(U_t) \;:=\; G^{-1}\big(U_t - c\big) \;=\; q_{s_t}(U_t), \ \forall s_t. \tag{31}$$

As a result, we conclude that

$$Z_t \;=\; \Phi_\theta\big(X_t; H_{t-1}\big) \;=\; q_{S_t}(U_t) \;=\; g\big(U_t\big) \quad \text{a.s..} \tag{32}$$

$\square$

**Corollary 4.5** (Counterfactual recovery). *Let Assumption 4.1 hold. Consider a factual sample generated by the structural causal model $X_t^{\text{F}} = f(X_{t-}, X_{\text{pa},t-}, U_t)$, and let its encoded latent be $Z_t^{\text{F}} := \Phi_\theta\big(X_t^{\text{F}}; H_{t-1}^{\text{F}}\big)$. At time step $t$, we obtain the counterfactual hidden state $\hat{H}_{t-1}^{\text{CF}}$ from the predicted counterfactual past $(\hat{X}_{t-}^{\text{CF}}, \hat{X}_{\text{pa},t-}^{\text{CF}})$. The decoder recovers the true counterfactual at time step $t$ almost surely:*

$$\hat{X}_t^{\text{CF}} := \Phi_\theta^{-1}\big(Z_t^{\text{F}}; \hat{H}_{t-1}^{\text{CF}}\big) = X_t^{\text{CF}}.$$

*Proof.* We establish the result by induction over time steps $t$. Following the definition of the true counterfactual value $X_t^{\text{CF}}$ defined in (15), we denote the corresponding true hidden state $H_{t-1}^{\text{CF}}$, which is updated using the true counterfactual values $X_{t-1}^{\text{CF}}$ and $X_{\text{pa},t-1}^{\text{CF}}$.

**Base case.** Let $t_0 \in \mathcal{I}$ be the first interventional step after the context window. By the intervention rule,

$$X_{t_0}^{\text{CF}} = \gamma_{t_0} \quad \text{and} \quad \hat{X}_{t_0}^{\text{CF}} = \gamma_{t_0},$$

hence $\hat{X}_{t_0}^{\text{CF}} = X_{t_0}^{\text{CF}}$. Moreover, since $t_0$ is the first intervention, the histories up to $t_0 - 1$ are factual in both constructions, so

$$\hat{H}_{t_0-1}^{\text{CF}} = H_{t_0-1}^{\text{CF}}.$$

**Induction step.** Suppose that for time step until $t-1$, the estimated counterfactual history matches the true one, i.e.,

$$\hat{X}_{t-}^{\mathrm{CF}} = X_{t-}^{\mathrm{CF}}.$$

Since the counterfactual hidden state is a function of past history until $t-1$, this implies

$$\hat{H}_{t-1}^{\mathrm{CF}} = H_{t-1}^{\mathrm{CF}}, \tag{33}$$

where the left-hand side is updated recursively from the estimated counterfactual history, and the right-hand side denotes the true hidden state under the true counterfactuals.

By the deterministic and invertible property of the flow for fixed conditioning input, we have

$$\Phi_\theta^{-1}\big(\Phi_\theta(X_t; H_{t-1});\ H_{t-1}\big) = X_t. \tag{34}$$

From the structural causal model, the factual sample is

$$X_t^{\mathrm{F}} = f(X_{t-}, X_{\mathrm{pa},t-}, U_t),$$

for some exogenous noise $U_t$. Fixing the same $U_t$, and under the intervention $\mathrm{do}(X_{t-} = X_{t-}^{\mathrm{CF}}, X_{\mathrm{pa},t-} = X_{\mathrm{pa},t-}^{\mathrm{CF}})$, the true counterfactual sample is

$$X_t^{\mathrm{CF}} = f(X_{t-}^{\mathrm{CF}}, X_{\mathrm{pa},t-}^{\mathrm{CF}}, U_t). \tag{35}$$

By Proposition 4.3, the latent factual representation satisfies

$$Z_t^{\mathrm{F}} = \Phi_\theta\big(X_t^{\mathrm{F}};\ H_{t-1}^{\mathrm{F}}\big) = \Phi_\theta\big(f(X_{t-}, X_{\mathrm{pa},t-}, U_t);\ H_{t-1}^{\mathrm{F}}\big) = g(U_t),$$

which depends only on $U_t$ and is invariant to $(X_{t-}, X_{\mathrm{pa},t-})$ and thus to $H_{t-1}$. Consequently, under the same intervention $\mathrm{do}(X_{t-} = X_{t-}^{\mathrm{CF}}, X_{\mathrm{pa},t-} = X_{\mathrm{pa},t-}^{\mathrm{CF}})$, we also have

$$\Phi_\theta\big(X_t^{\mathrm{F}};\ H_{t-1}^{\mathrm{F}}\big) = \Phi_\theta\big(f(X_{t-}, X_{\mathrm{pa},t-}, U_t); H_{t-1}^{\mathrm{F}}\big) = g\big(U_t\big) = \Phi_\theta\big(f(X_{t-}^{\mathrm{CF}}, X_{\mathrm{pa},t-}^{\mathrm{CF}}, U_t); H_{t-1}^{\mathrm{CF}}\big) = \Phi_\theta(X_t^{\mathrm{CF}}; H_{t-1}^{\mathrm{CF}}), \tag{36}$$

and more simply, we have:

$$\Phi_\theta\big(X_t^{\mathrm{F}};\ H_{t-1}^{\mathrm{F}}\big) = \Phi_\theta(X_t^{\mathrm{CF}}; H_{t-1}^{\mathrm{CF}}). \tag{37}$$

Combining (33) and (37), we obtain

$$\Phi_\theta^{-1}\big(\Phi_\theta(X_t^{\mathrm{F}}; H_{t-1}^{\mathrm{F}});\ \hat{H}_{t-1}^{\mathrm{CF}}\big) = \Phi_\theta^{-1}\big(\Phi_\theta(X_t^{\mathrm{CF}}; H_{t-1}^{\mathrm{CF}});\ H_{t-1}^{\mathrm{CF}}\big). \tag{38}$$

By (34), the right-hand side of (38) equals $X_t^{\mathrm{CF}}$. Since the left-hand side of (38) is precisely the algorithm's counterfactual encoder-decoder procedure, it follows that

$$\hat{X}_t^{\mathrm{CF}} := \Phi_\theta^{-1}\big(Z_t^{\mathrm{F}};\ \hat{H}_{t-1}^{\mathrm{CF}}\big) = X_t^{\mathrm{CF}}.$$

Therefore, the decoder recovers the true counterfactual at time step $t$. By induction, the claim holds for all $t$. $\qquad\square$

# B  CAUSAL SCM ASSUMPTIONS AND THEORETICAL COMPARISONS TO BGM

## B.1  CAUSAL SCM ASSUMPTIONS

Let $\mathbf{X}_{\tau+1:T}^{(\gamma_\mathcal{I})}$ denote the potential trajectory under the intervention $\mathrm{do}(X_\mathcal{I} := \gamma_\mathcal{I})$, where $\mathcal{I} \subseteq [K] \times \{\tau+1, \ldots, T\}$. We assume the following standard conditions for the structural causal model on the given DAG:

**Assumption B.1.**
(1) *Unconfoundedness.* The exogenous noises $(U_{i,t})$ are mutually independent across $(i, t)$, and all common causes of observed nodes are included in the DAG, i.e., no unobserved confounders for the interventions considered.
(2) *Consistency.* If the system is run under $\mathrm{do}(X_\mathcal{I} := \gamma_\mathcal{I})$, then the observed trajectory equals the corresponding potential trajectory under the regime actually applied: $\mathbf{X}_{\tau+1:T} = \mathbf{X}_{\tau+1:T}^{(\gamma_\mathcal{I})}$.

(3) *Positivity.* For any $(x_{i,t-}, x_{\mathrm{pa}(i),t-})$ in the support of $(X_{i,t-}, X_{\mathrm{pa}(i),t-})$, the conditional density of $X_{i,t}$ given these parents is positive in a neighborhood of the intervention value $\gamma_{i,t}$:

$$p\big(X_{i,t} = x \,\big|\, X_{i,t-} = x_{i,t-}, X_{\mathrm{pa}(i),t-} = x_{\mathrm{pa}(i),t-}\big) > 0.$$

(4) *No Interference.* For any two intervention schedules $\mathcal{I}$ and $\mathcal{J}$ with the same assigned values on a given index set, the corresponding potential trajectories coincide on that set. Interventions affect other nodes only through the edges of the DAG.

**Remark B.2.** Assumption B.1 (1) imposes causal sufficiency: all common causes of the observed nodes are included in the DAG, so the exogenous noises $(U_{i,t})$ are mutually independent across $(i,t)$. This rules out time–varying latent confounders that simultaneously influence multiple series. In settings where unobserved confounding is suspected, our identification and recovery guarantees for interventional and counterfactual distributions need not hold.

Extending DoFlow beyond the causal sufficiency assumption to settings with latent confounders is an important direction for future work. Recent causal generative frameworks such as DeCaFlow (Almodóvar et al., 2025) and Neural Causal Models (Xia et al., 2022) explicitly address hidden confounding problems. DeCaFlow augments causal normalizing flows with latent confounders and proxy variables to recover identifiable interventional and counterfactual queries. Besides, Neural Causal Models use neural SCMs with correlated exogenous variables and graphical constraints to decide and estimate identifiable counterfactuals from observational data. A promising direction is to combine such deconfounding mechanisms with our time-indexed CNF parameterization so that DoFlow can handle deconfounded temporal DAG.

## B.2 Theoretical Comparisons to BGM

**Bijective Generation Mechanisms (BGM).** The BGM framework (Nasr-Esfahany et al., 2023) shows that if the true structural mechanism $f$ is in the BGM class (bijective/strictly monotone in the exogenous noise), then any learned mechanism $\hat{f}$ that (i) is also in the BGM class and (ii) *matches the observed distribution*—i.e., for the same parents value $X$ and action $A$,

$$\hat{f}(X_{t-}, X_{\mathrm{pa},t-}, U_t) \overset{d}{=} f(X_{t-}, X_{\mathrm{pa},t-}, U_t)$$
$$i.e., (\hat{f}(X_{t-}, X_{\mathrm{pa},t-}, \cdot))_{\#} P_U = (f(X_{t-}, X_{\mathrm{pa},t-}, \cdot))_{\#} P_U,$$

yields the *same counterfactuals* as $f$. Their result is model-agnostic, at the class level.

**DoFlow.** Our findings are specific to a particular methodology and architecture, as this is a methodology paper where we analyze continuous normalizing flows (CNFs). With the additional assumption (A3) specific to DoFlow, Proposition 4.3 shows that the DoFlow's encoded latent $Z_t$ is bijective in the exogenous noise $U_t$. Consequently, Corollary 4.5 proves that the encode–decode procedure recovers the true counterfactual.

**Relationships.** Because of Proposition 4.3 (enabled by the CNF), DoFlow implements a bijective-in-noise mechanism and is thus comparable to BGM in the sense that $\hat{f}$ is bijective in $U_t$. However, they are not the same: BGM additionally requires observational distribution matching (as formalized in (A4)) to obtain class-level identifiability, whereas our proofs of Proposition 4.3 and Corollary 4.5 do not assume (A4) and instead rely on the model-specific condition (A3).

**Alternative route.** The counterfactual recovery result (Corollary 4.5) can alternatively be established by imposing an additional assumption:

**(A4) Observational matching.** For each $(X_{t-}, X_{\mathrm{pa},t-})$, the DoFlow–induced observational law matches the true one, i.e.

$$(\hat{f}(X_{t-}, X_{\mathrm{pa},t-}, \cdot))_{\#} P_U = (f(X_{t-}, X_{\mathrm{pa},t-}, \cdot))_{\#} P_U.$$

Under (A1)–(A3), Proposition 4.3 establishes a continuously differentiable bijection $g : U_t \to Z_t$ (independent of $(X_{t-}, X_{\mathrm{pa},t-})$), so the induced mechanism $\hat{f}$ implemented by DoFlow is bijective in the exogenous noise, i.e., DoFlow lies in the BGM class. With the additional assumption (A4), we can therefore directly invoke the counterfactual recovery result of the BGM framework (Nasr-Esfahany et al., 2023).

## C   PRELIMINARIES ON CONTINUOUS NORMALIZING FLOWS

For better logical flow in the main text, we move the CNF preliminaries here. Continuous normalizing flows (CNFs) have been successful for statistical sampling (Tian et al., 2024; Wu and Xie, 2025), structural generation (Guo et al., 2025; Wu et al., 2026), and counterfactual inference (Wu et al., 2025; Chao et al., 2024). This appendix section reviews CNF fundamentals and flow-matching training for general data types.

### C.1   NEURAL ODE AND CONTINUOUS NORMALIZING FLOW:

A Neural ODE models the evolution of a sample as the solution to an ordinary differential equation (ODE). Concretely, in $\mathbb{R}^d$, given an initial condition $x_0 = x(0)$ at $s = 0$, the transformation to the output $x_1 = x(1)$ at $s = 1$ is governed by:

$$\frac{dx(s)}{ds} = v(x(s), s), \quad s \in [0, 1], \tag{39}$$

where $v : \mathbb{R}^d \times [0, 1] \to \mathbb{R}^d$ is the velocity field parameterized by a neural network. The time horizon is rescaled to $s \in [0, 1]$ without loss of generality.

Continuous Normalizing Flow (CNF) is a class of normalizing flows in which the transformation of a probability density is governed by a time-continuous Neural ODE. Let $p(x, s)$ denote the marginal density of $x(s)$. Then $p(x, s)$ evolves according to the Liouville continuity equation implied by (39):

$$\partial_s p(x, s) + \nabla \cdot \big( p(x, s) \, v(x, s) \big) = 0, \quad s \in [0, 1], \tag{40}$$

where $\nabla \cdot$ denotes the divergence operator.

When the Neural ODE is well-posed, it induces a continuous and invertible map from the initial sample $x_0$ to the terminal sample $x_1$. The inverse map is obtained by integrating (39) backward in time. This mechanism allows one to choose $x_0 \sim p(\cdot, 0)$ as the data distribution and $x_1 \sim p(\cdot, 1)$ as a simple base (noise) distribution, typically $N(0, I)$. For convenience, we write $q(\cdot) := p(\cdot, 1)$ for the base distribution. Throughout, we use the common choice $q(\cdot) = N(0, I)$.

### C.2   FLOW MATCHING

Flow Matching (FM) (Lipman et al., 2023) trains continuous normalizing flows without simulating trajectories by regressing the model velocity $v(x(s), s)$ toward a prescribed target field $u(x(s), s)$. The FM objective is

$$\mathcal{L}_{\text{FM}} = \mathbb{E}_{s \sim \mathcal{U}[0,1], \, x \sim p(\cdot, s)} \left[ \| v(x(s), s) - u(x(s), s) \|^2 \right], \tag{41}$$

where $u(x(s), s)$ is analytically specified.

**Linear Interpolant**: Because $u(x, s)$ is generally intractable unless we condition on the starting point $x_0$, Conditional Flow Matching (CFM) (Lipman et al., 2023) was proposed to find a tractable velocity. Given an observed data $x_0$ and an endpoint $x_1 \sim q(\cdot)$ from base distribution, we can choose an analytic interpolation between $x_0$ and $x_1$, and define a reference path. An information-preserving and simple choice is the linear interpolant (Lipman et al., 2023):

$$\phi(x_0, x_1; s) = (1 - s)x_0 + (s + \sigma_{\min}(1 - s))x_1, \tag{42}$$

where $x_1 \sim N(0, I)$ and $\sigma_{\min}$ is a small positive hyperparameter ensuring $p(\phi, 0) \sim N(x_0, \sigma_{\min}^2 I)$. Setting $\sigma_{\min} = 0$ recovers the strict linear path $\phi = (1 - s)x_0 + sx_1$.

We treat $\phi$ as a fixed reference trajectory that the learned flow is trained to track. Under (42), the associated reference velocity is

$$\frac{d\phi}{ds} = (1 - \sigma_{\min})x_1 - x_0. \tag{43}$$

**Training loss**: CFM trains the flow $v(x(s), s)$ by directly regressing it onto the reference velocity field (43). The training objective is given by:

$$\mathcal{L}_{\text{CFM}} = \mathbb{E}_{s \sim \mathcal{U}[0,1], x_0 \sim p(\cdot, 0), x_1 \sim q(\cdot)} \| v(\phi, s) - \frac{d\phi}{ds} \|^2. \tag{44}$$

# D  DATA SYNTHESIS AND METRICS DEFINITION

We define four types of causal DAG structures: Tree, Diamond, Fully Connected Layer (FC-Layer), and Chain with skip connections (Chain). For each structure, we design both additive models and nonlinear, non-additive structural causal models.

In the simulations for different graph structures, the root node $X_1$ (for Tree, Diamond, and Chain), and the root nodes $X_1, X_2, X_3$ (for FC-Layer) are initialized over the interval $[0, t_0]$ using a Chain process:

$$X_{i,t} = \beta_1 X_{i,t-1} + A \sin\left(\frac{2\pi t}{P} + \phi\right) + U_{i,t}. \tag{45}$$

Here, $i \in \{1\}$ for Tree, Diamond, and Chain, and $i \in \{1, 2, 3\}$ for FC-Layer. Besides, $A$, $P$, and $\phi_i$ denote the amplitude, period, and phase offset of the sinusoidal driver, and $U_{i,t}$ represents independent exogenous noise $N(0, 1)$.

In the Tree, Diamond, and Chain structures, the hyperparameters for the root node $X_{1,t}$- namely $A$, $P$, and $\phi$, are set to 1.0, 20, and $\frac{2\pi}{9}$, respectively. For the FC-Layer, these are set to 1.5, 15, and $\frac{4\pi}{9}$, respectively.

## D.1  TREE

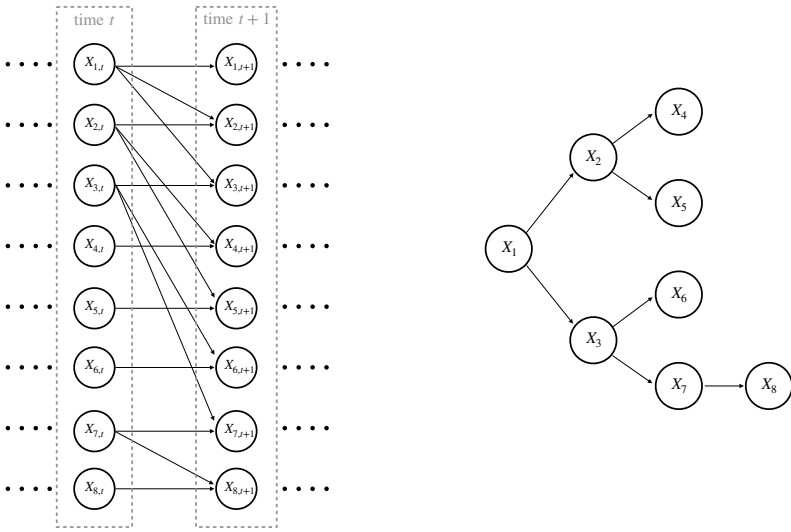

Figure 4: **Tree** graph over 8 nodes. Exogenous variables $U_{i,t}$ are omitted for clarity but exist for every node at each time $t$. **Left:** Full node-level causal structure between consecutive time, with all variables $\{X_{1,t}, \ldots, X_{8,t}\}$ present at each step. **Right:** Rolled-up (time-suppressed) view over different nodes $\{X_1, \ldots, X_8\}$. Each arrow $X_i \to X_j$ (with $i \neq j$) denotes a lag-1 temporal dependency $X_{i,t-1} \to X_{j,t}$ that holds for all $t$. Both panels depict the same underlying structure.

We consider both additive and nonlinear, non-additive structural causal models:

- **Additive model:**
$$X_{i,t} = f(X_{i,t-1}, X_{\mathrm{pa}(i),t-1}, U_{i,t}) = \beta_i X_{i,t-1} + \sum_{j \in \mathrm{pa}(i)} \tilde{\beta}_{i \leftarrow j} X_{j,t-1} + U_{i,t}/4. \tag{46}$$

- **Nonlinear, non-additive model:**
$$X_{i,t} = f(X_{i,t-1}, X_{\mathrm{pa}(i),t-1}, U_{i,t}) = \beta_i X_{i,t-1}\left(|U_{i,t}| + 0.5\right) + \sum_{j \in \mathrm{pa}(i)} \tilde{\beta}_{i \leftarrow j} X_{j,t-1}. \tag{47}$$

For the Tree graph, we set the self-lag coefficients $\beta_i \in \{0.5, 0.4, 0.3, 0.2, 0.1, 0.2, 0.2, 0.2\}$ for $i = 1, \ldots, 8$ (in order), and define edge-specific parent coefficients $\tilde{\beta}_{i \leftarrow j}$ only on the directed edges $1 \to 2$ (0.3), $1 \to 3$ (-0.3), $2 \to 4$ (0.3), $2 \to 5$ (-0.3), $3 \to 6$ (0.5), $3 \to 7$ (-0.5), and $7 \to 8$ (0.5), with $\tilde{\beta}_{i \leftarrow j} = 0$ for all other pairs.

## D.2 DIAMOND

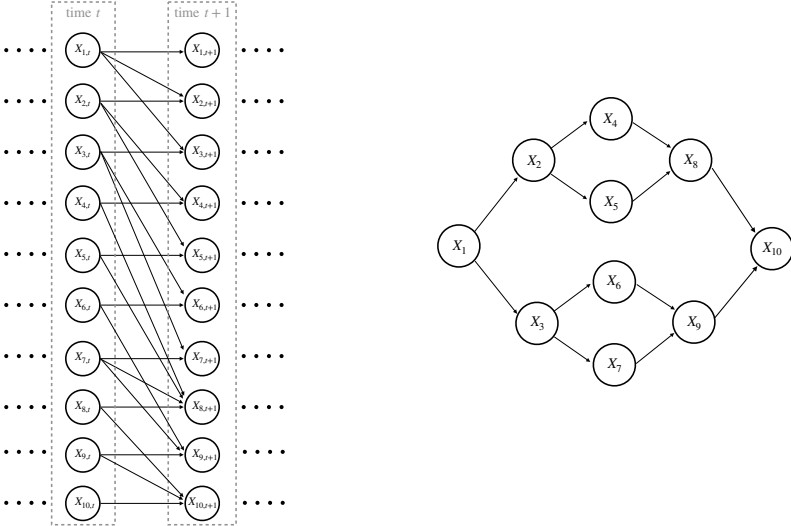

Figure 5: **Diamond** graph over 10 nodes. Exogenous variables $U_{i,t}$ are omitted for clarity but exist for every node at each time $t$. **Left:** Full node-level causal structure between consecutive time, with all variables $\{X_{1,t}, \ldots, X_{10,t}\}$ present at each step. **Right:** Rolled-up (time-suppressed) view over different nodes $\{X_1, \ldots, X_{10}\}$. Each arrow $X_i \to X_j$ (with $i \neq j$) denotes a lag-1 temporal dependency $X_{i,t-1} \to X_{j,t}$ that holds for all $t$. Both panels depict the same underlying structure.

We consider both additive and nonlinear, non-additive structural causal models:

- **Additive model:**

$$X_{i,t} = f(X_{i,t-1}, X_{\mathrm{pa}(i),t-1}, U_{i,t}) = \beta_i X_{i,t-1} + \sum_{j \in \mathrm{pa}(i)} \tilde{\beta}_{i \leftarrow j} X_{j,t-1} + U_{i,t}. \quad (48)$$

- **Nonlinear, non-additive model:**

$$X_{i,t} = f(X_{i,t-1}, X_{\mathrm{pa}(i),t-1}, U_{i,t}) = \exp(\beta_i X_{i,t-1}) \cdot \frac{1}{2 + |U_{i,t}|} + \sum_{j \in \mathrm{pa}(i)} \tilde{\beta}_{i \leftarrow j} X_{j,t-1}. \quad (49)$$

For the Diamond graph, we set the self-lag coefficients $(\beta_1, \ldots, \beta_{10}) = (0.5, 0.4, 0.3, 0.2, 0.1, 0.2, 0.2, 0.2, 0.2, 0.2)$, and define edge-specific parent coefficients $\tilde{\beta}_{i \leftarrow j}$ only on the directed edges $1 \to 2$ (0.3), $1 \to 3$ (0.3), $2 \to 4$ (0.3), $2 \to 6$ (0.3), $3 \to 5$ (0.5), $3 \to 7$ (0.5), $4 \to 8$ (0.5), $6 \to 8$ (0.5), $5 \to 9$ (0.5), $7 \to 9$ (0.5), $8 \to 10$ (0.5), and $9 \to 10$ (0.5), with $\tilde{\beta}_{i \leftarrow j} = 0$ for all other pairs.

## D.3 FULLY CONNECTED LAYER (FC-LAYER)

We consider both additive and nonlinear, non-additive structural causal models:

- **Additive**

$$X_{i,t} = f(X_{i,t-1}, X_{\mathrm{pa}(i),t-1}, U_{i,t}) = \beta_i X_{i,t-1} + \sum_{j \in \mathrm{pa}(i)} \tilde{\beta}_{i \leftarrow j} X_{j,t-1} + U_{i,t}. \quad (50)$$

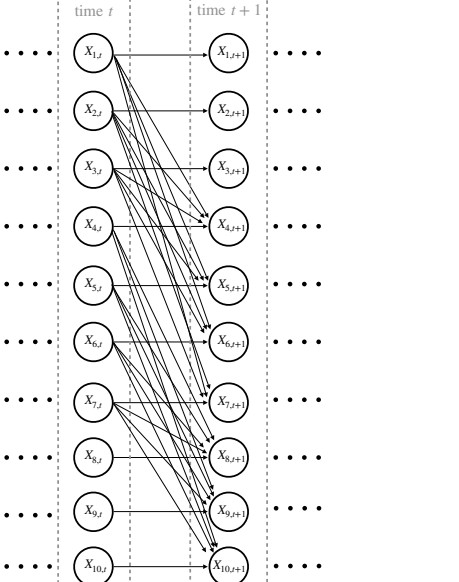 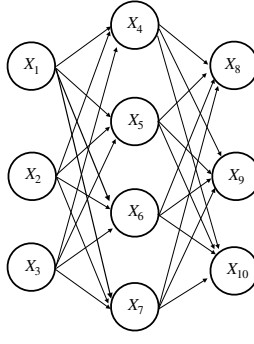

Figure 6: **FC-Layer** graph over 10 nodes. Exogenous variables $U_{i,t}$ are omitted for clarity but exist for every node at each time $t$. **Left:** Full node-level causal structure between consecutive time, with all variables $\{X_{1,t}, \ldots, X_{10,t}\}$ present at each step. **Right:** Rolled-up (time-suppressed) view over different nodes $\{X_1, \ldots, X_{10}\}$. Each arrow $X_i \rightarrow X_j$ (with $i \neq j$) denotes a lag-1 temporal dependency $X_{i,t-1} \rightarrow X_{j,t}$ that holds for all $t$. Both panels depict the same underlying structure.

- **Nonlinear, non-additive model:**

$$X_{i,t} = f(X_{i,t-1}, X_{\mathrm{pa}(i),t-1}, U_{i,t}) = \beta_i X_{i,t-1} + \left( \sum_{j \in \mathrm{pa}(i)} \tilde{\beta}_{i \leftarrow j} X_{j,t-1} \right) +$$

$$\gamma \left( \exp\Big( \sum_{j \in \mathrm{pa}(i)} \tilde{\beta}_{i \leftarrow j} X_{j,t-1} - 1 \Big) - 1 \right) - \gamma \tanh\Big( \sum_{j \in \mathrm{pa}(i)} \tilde{\beta}_{i \leftarrow j} X_{j,t-1} \Big) + U_{i,t}. \tag{51}$$

For the FC-Layer graph, we set $\gamma = 0.30$ and the self-lag coefficients $(\beta_1, \ldots, \beta_{10}) = (0.5, 0.6, 0.7, 0.4, 0.45, 0.5, 0.55, 0.3, 0.35, 0.4)$. We define edge-specific parent coefficients $\tilde{\beta}_{i \leftarrow j}$ densely between adjacent layers: for every hidden node $i \in \{4, 5, 6, 7\}$ and input $j \in \{1, 2, 3\}$, we set $\tilde{\beta}_{i \leftarrow j} = 0.2 + 0.1\,(j - 1)$ (i.e., $0.2, 0.3, 0.4$ for $j = 1, 2, 3$), and for every output node $i \in \{8, 9, 10\}$ and hidden $j \in \{4, 5, 6, 7\}$ we set $\tilde{\beta}_{i \leftarrow j} = 0.2$, with $\tilde{\beta}_{i \leftarrow j} = 0$ otherwise.

### D.4 CHAIN LINEAR (CHAIN)

We consider both additive and nonlinear, non-additive structural causal models:

- **Additive model:**

$$X_{i,t} = f(X_{i,t-1}, X_{\mathrm{pa}(i),t-1}, U_{i,t}) = \beta_i X_{i,t-1} + \sum_{j \in \mathrm{pa}(i)} \tilde{\beta}_{i \leftarrow j} X_{j,t-1} + U_{i,t}. \tag{52}$$

- **Nonlinear, non-additive model:**

$$X_{i,t} = f(X_{i,t-1}, X_{\mathrm{pa}(i),t-1}, U_{i,t}) = \big(\beta_i X_{i,t-1} + 0.5\big)\,|U_{i,t}| + \tilde{\beta}_{i \leftarrow i-1} X_{i-1,t-1} + \tilde{\beta}_{i \leftarrow 1} X_{1,t-1} \tag{53}$$

For the Chain structure, we set $\beta_i = 0.6$ for all $i = 1, \ldots, 50$, and draw independent Gaussian noises $U_{i,t} \sim \mathcal{N}(0, \sigma_i^2)$ with $\sigma_i = 0.2$. For each non-root node $i = 2, \ldots, 50$, the immediate-predecessor coefficient is fixed as $\tilde{\beta}_{i \leftarrow i-1} = 0.2$, and the skip connection from the root is fixed as $\tilde{\beta}_{i \leftarrow 1} = 0.7$.

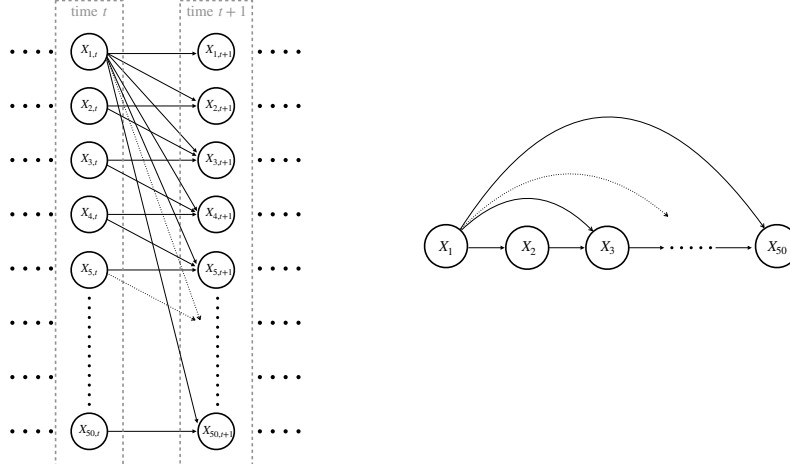

Figure 7: **Chain** graph over 50 nodes. Exogenous variables $U_{i,t}$ are omitted for clarity but exist for every node at each time $t$. **Left:** Full node-level causal structure between consecutive time, with all variables $\{X_{1,t}, \ldots, X_{50,t}\}$ present at each step. **Right:** Rolled-up (time-suppressed) view over different nodes $\{X_1, \ldots, X_{50}\}$. Each arrow $X_i \to X_j$ (with $i \neq j$) denotes a lag-1 temporal dependency $X_{i,t-1} \to X_{j,t}$ that holds for all $t$. Both panels depict the same underlying structure.

## D.5 INTERVENTIONAL AND COUNTERFACTUAL SIMULATIONS

Each time series window has length $T = 120$. We set the context window to $\tau = 90$ and the forecasting window to $T - \tau = 30$. Interventions begin immediately after the context window $\{1, 2, \ldots, \tau\}$.

To rigorously evaluate DoFlow under challenging causal interventions, we design interventions that alter the root node(s) of the causal DAG, since manipulating root causes propagates system-wide effects through all downstream variables. Each root node follows a cyclic structural equation (45).

To ensure that the intervention produces a fundamentally different dynamic pattern, we construct the interventional trajectory by phase-shifting the root cycle by half a period, i.e., $P/2$. This yields:

$$\gamma_{i,t} = \beta_1 \gamma_{i,t-1} + A\sin\left(\tfrac{2\pi(t+P/2)}{P} + \phi\right) + U_{i,t}, \quad t \in \{\tau+1, \ldots, T\} \tag{54}$$

where $i \in \{1\}$ for Tree, Diamond, and Chain, and $i \in \{1, 2, 3\}$ for FC-Layer. This corresponds to a complete inversion of the cyclic trend for the root nodes of each graph structure (including a noise term to create different intervention schedules across runs). Consequently, the interventional trajectory exhibits an antiphase pattern relative to the original, creating the most challenging regime for causal forecasting and counterfactual recovery.

Below, we illustrate how to simulate the true interventional and counterfactual trajectories using the SCMs defined above. For clarity, we present the procedure for the Chain, Diamond, and Tree structures, where the intervened root node is $X_{1,t}$. The FC-Layer case follows identically, except that interventions are applied to root nodes $i \in \{1, 2, 3\}$ as defined in (54).

The interventional simulation proceeds by first sampling the exogenous noises $\{U_{i,\tau+1,T}\}_{i=2}^{K}$. The intervened values are then obtained by applying $\mathrm{do}(X_{1,\tau+1:T} = \gamma_{1,\tau+1:T})$ and propagating forward with the sampled noises to generate the intervened trajectories $\{\tilde{X}_{i,\tau+1:T}\}_{i=2}^{K}$. For the FC-Layer graph, the intervened parental set includes $\{X_{i,t}\}_{i=1,2,3}$.

Counterfactual simulation begins by recovering the exogenous noises $\{U_{i,t}\}_{i>1, t\in[\tau+1,T]}$ from the observed factual future $\{X_{i,\tau+1:T}^{\mathrm{F}}\}_{i=2}^{K}$ using the structural causal models. Next, the intervention $\mathrm{do}(X_{1,\tau+1,T} = \gamma_{1,\tau+1:T})$ is applied, and the system is propagated forward with the recovered noises to generate the counterfactual trajectories $\{X_{i,\tau+1:T}^{\mathrm{CF}}\}_{i=2}^{K}$. For the FC-Layer graph, the intervened parental set includes $\{X_{i,t}\}_{i=1,2,3}$.

## D.6 METRICS

We evaluate model performance using both Root Mean Squared Error (RMSE) and Maximum Mean Discrepancy (MMD). Let the test batch size be $B = 128$, the ground-truth value for node $i$ at time $t$ in batch $b$ be $x_{i,t}^{(b)}$, and the corresponding model prediction be $\hat{x}_{i,t}^{(b)}$.

For each test sequence $b$, given the same context $\{x_{i,1:\tau}^{(b)}\}_{i=1}^K$, we generate $N = 100$ realizations of both the model-estimated and the true observational/interventional forecasting trajectories, in order to obtain more accurate evaluation metrics.

To ensure comparability across different scales of time series, we apply standard normal scaling to each batch $b$ over the forecasting window, using the mean $\mu_i^{(b)}$ and standard deviation $\sigma_i^{(b)}$ computed from its context window of node $i$. The generated time series dataset has a total length of 20,000, with a stride of 1, resulting in 15,881 training samples (80%) and 3,881 testing samples (20%).

### D.6.1 RMSE

The RMSE of node $i$ for a single realization is defined as

$$\text{RMSE}_i = \sqrt{\frac{1}{(T-\tau)B} \sum_{t=\tau+1}^{T} \sum_{b=1}^{B} \big(\hat{x}_{i,t}^{(b)} - x_{i,t}^{(b)}\big)^2}. \tag{55}$$

For observational and interventional testing, we evaluate each method over 50 test batches of batch size $B = 128$. Within a batch, RMSE is computed by (i) averaging over nodes $i = 1, \ldots, K$, then (ii) averaging over $N = 100$ stochastic realizations, and finally (iii) averaging over the 50 batches. The reported standard deviation (std) reflects variation across the 50 batches.

For counterfactual testing, each method produces a single deterministic counterfactual trajectory for a given test instance, so $N = 1$. We evaluate performance by averaging RMSE over counterfactual test batches (each with $B = 128$), and report std across 50 batches.

### D.6.2 MMD

To calculate MMD, we first flatten each trajectory (length $T - \tau$, dimension $D$) into a vector in $\mathbb{R}^{(T-\tau)D}$. The sample size of both the true and the estimated trajectories $\{x_{i,\tau+1:T}\}_{i=1}^K$ is $BN$, since for each batch $b$ we simulate $N = 100$ realizations. The empirical MMD is then defined as

$$\widehat{\text{MMD}}^2 = \frac{1}{(BN)(BN-1)} \sum_{\substack{a,a'=1 \\ a \neq a'}}^{BN} k(x_a, x_{a'}) + \frac{1}{(BN)(BN-1)} \sum_{\substack{b,b'=1 \\ b \neq b'}}^{BN} k(\tilde{x}_b, \tilde{x}_{b'}) - \frac{2}{(BN)^2} \sum_{a=1}^{BN} \sum_{b=1}^{BN} k(x_a, \tilde{x}_b),$$

$$\tag{56}$$

where we use the Gaussian kernel

$$k(x, x') = \exp\Big(-\frac{\|x-x'\|^2}{2\sigma^2}\Big),$$

and the bandwidth $\sigma$ is chosen via the pooled median heuristic.

### D.6.3 CRPS

For probabilistic forecasting, we evaluate the quality of the predictive distribution using the Continuous Ranked Probability Score (CRPS) (Matheson and Winkler, 1976). For a univariate predictive CDF $F$ and a realized outcome $y \in \mathbb{R}$, the CRPS is defined as

$$\text{CRPS}(F, y) := \int_{-\infty}^{\infty} \Big(F(z) - \mathbf{1}\{z \geq y\}\Big)^2 dz, \tag{57}$$

where $\mathbf{1}\{\cdot\}$ is the indicator function.

In our experiments, $F$ is represented by an ensemble of samples $\{\hat{x}_{i,t}^{(b,n)}\}_{n=1}^N$ for node $i$, time $t$, and test sequence $b$, with realized value $x_{i,t}^{(b)}$. We use the standard ensemble approximation:

$$\text{CRPS}_{i,t}^{(b)} \approx \frac{1}{N} \sum_{n=1}^{N} |\hat{x}_{i,t}^{(b,n)} - x_{i,t}^{(b)}| - \frac{1}{2N^2} \sum_{n=1}^{N} \sum_{n'=1}^{N} |\hat{x}_{i,t}^{(b,n)} - \hat{x}_{i,t}^{(b,n')}|. \tag{58}$$

We report CRPS averaged over all test sequences $b$, time steps $t \in \{\tau+1, \ldots, T\}$, nodes $i = 1, \ldots, K$, and runs, analogous to the aggregation used for RMSE.

Notably, CRPS is applicable only to probabilistic models. Therefore, we report CRPS results only for DoFlow, DeepVAR, and MQF2.

## E    MODEL AND IMPLEMENTATION DETAILS

This section summarizes the main implementation and hyperparameter choices used in all experiments.

**Common setup.**    All models are implemented in PyTorch, with continuous normalizing flows (CNFs) integrated using `torchdiffeq.odeint`. Unless otherwise stated, we use: Adam optimizer with learning rate $10^{-3}$, batch size 128, and ReLU activations in the CNF networks. For each node $i$ in the DAG, we instantiate for our algorithm: (i) an LSTM-based RNN encoder that takes as input the node value (and, when applicable, time features), and (ii) a CNF whose velocity field is parameterized by a 3-layer MLP with hidden width 64. All time series are normalized per window using a mean–variance scaler over the context window.

### E.1    ADAPTING OBSERVATIONAL BASELINES FOR INTERVENTIONAL ROLLOUTS

For each baseline method, we enforce the causal factorization by training a separate model for each node $i$, using as input a rolling context window consisting of the node history and its parents' histories, $\{X_{i,1:t-1}, X_{\mathrm{pa}(i),1:t-1}\}$, and predicting the next-step target $X_{i,t}$. For architectures that natively support multi-horizon prediction, we use the same model class but set the prediction length to one step and apply it recursively.

To predict interventional future under an intervention schedule $\mathcal{I}$ with values $\{\gamma_{i,t}\}$, we perform multi-step rollout over the forecast window by iterating $t = \tau + 1, \ldots, T$ and forecasting nodes in topological order: if $(i, t) \in \mathcal{I}$ we fix $\hat{x}_{i,t} = \gamma_{i,t}$; otherwise we predict $\hat{x}_{i,t}$ from the baseline model using the latest rolled-out histories. The rolled-out predictions (including fixed intervention values) are then fed back as inputs for subsequent time steps, ensuring that interventions propagate to downstream nodes through the DAG.

### E.2    SYNTHETIC SCM EXPERIMENTS

For the synthetic DAG experiments (Tree, Diamond, Chain, and FC-Layer graphs), we first generate long multivariate time series using the SCMs in Appendix D. We then extract sliding windows for training and evaluation.

Each window has total length $T = 120$, with a context window of $\tau = 90$ steps and a forecasting horizon of $T-\tau = 30$ steps. We construct windows with stride 1 along the simulated sequence. The resulting windows are split along the time index into 80% for training and 20% for testing, using the first $0.8N$ time stamps for training and the remaining $0.2N$ for testing.

For all synthetic structures, we set up our algorithm:

- Per-node RNN encoder: 3-layer LSTM with hidden size 15.
- Per-node CNF: MLP with hidden dimension 64 and 3 layers; the conditioning dimension is

$$\mathrm{cond\_dim}_i = 15 + |\mathrm{pa}(i)| \cdot 15.$$

We train DoFlow for 15 epochs on the synthetic datasets. For interventional evaluation, we draw 100 flow samples per window to plot the confidence interval as well as to calculate the metrics.

### E.3    HYDROPOWER FORECASTING

For the hydropower case study, we use processed Argonne hydropower data at individual locations. From the raw time-stamped series, we extract windows of length

$$\mathrm{context\_length} = 180, \quad \mathrm{prediction\_length} = 30,$$

with stride 5 along time. Thus each training and test window contains 180 context time steps followed by a 30-step forecasting horizon.

We incorporate the time features, including the time of day and calendar features, to obtain a time-feature dimension of TF = 6. We set up our algorithm:

- Per-node RNN encoder: 3-layer LSTM with hidden size 15.
- Per-node CNF: MLP with hidden dimension 64 and 3 layers; the conditioning dimension is

$$\text{cond\_dim}_i = 15 + |\text{pa}(i)| \cdot 15,$$

including the node's own hidden state, and the hidden states of its parents.

For training (performed offline and loaded from checkpoints in the main hydropower experiments), we use batch size 128, Adam with learning rate $10^{-3}$, and $N_{\text{epochs}} = 15$ with early stopping based on node-wise training loss.

### E.4 CANCER TREATMENT EFFECT

For the cancer-treatment experiment, we use the public dataset from Bica et al. (2020a), preprocessed into patient-level sequences with actions and outcomes. We construct per-patient windows of length

$$\text{context\_len} = 55, \quad \text{pred\_len} = 5,$$

where the context contains the past tumor volumes and treatments, and the prediction horizon contains the next 5 steps to be modeled under different treatment policies.

Here we distinguish between outcome and action histories via two separate RNNs in our algorithm:

- Outcome RNN: 4-layer LSTM with hidden size $H = 15$ and input size 1 (normalized cancer volume).
- Action RNN: 4-layer LSTM with hidden size $H = 15$ and input size $D_A = 4$ (chemo application, radio application, chemo dose, radio dose).

The CNF for the cancer outcome $Y_t$ is parameterized as a 3-layer MLP with hidden width 64, with conditioning dimension

$$\text{cond\_dim} = 15 + 15,$$

corresponding to the concatenation of the current outcome hidden state and action hidden state.

We construct a training set of patient windows via `CancerTrainDataset` with batch size 128 and shuffle the patients at each epoch. We train the cancer-treatment model for 15 epochs, using the same per-window mean–variance normalization as in the other experiments. All reported interventional and counterfactual metrics are computed by rolling out DoFlow over the 5-step prediction horizon under alternative (simulated) treatment policies, as detailed in Section F.4.

## F MORE EXPERIMENTAL RESULTS

### F.1 COMPUTATIONAL COSTS

The number of training samples is 15,881 for each simulated datasets. We use a batch size of 128, and all experiments—including training and sampling time comparisons—are conducted on a single A100 GPU.

Table 2: Comparison of model size.

|  | Tree | Diamond | Layer | Chain | Hydropower |
|---|---|---|---|---|---|
| **DoFlow** | $94,664$ | $121,658$ | $133,946$ | $646,178$ | $99,592$ |
| GRU | $36,830$ | $50,106$ | $51,732$ | $124,412$ | $52,026$ |
| TFT | $94,860$ | $100,472$ | $124,775$ | $531,026$ | $95,318$ |
| TiDE | $108,301$ | $118,332$ | $118,332$ | $544,539$ | $122,462$ |
| TSMixer | $117,818$ | $119,318$ | $119,318$ | $550,177$ | $110,518$ |
| DeepVAR | $73,712$ | $89,280$ | $89,280$ | $184,280$ | $84,290$ |
| MQF2 | $128,677$ | $160,846$ | $160,846$ | $702,681$ | $125,459$ |

Table 3: Comparison of training time per epoch, epochs to convergence, and *observational* sampling time (for 1,000 forecast series over a 30-step horizon) across Tree, Diamond, and Layer structures.

| Methods | Training Time / Epoch | Epochs to Conv. | Total Training Time | Sampling Time |
|---|---|---|---|---|
| **DoFlow** | 42.3s | 10 | 8.09min | 10.25s |
| GRU | 17.8s | 15 | 5.03min | 6.80s |
| TFT | 21.0s | 15 | 5.28min | 11.07s |
| TiDE | 29.3s | 25 | 12.1min | 2.23s |
| TSMixer | 27.9s | 25 | 12.1min | 8.83s |
| DeepVAR | 20.6s | 20 | 11.65min | 7.86s |
| MQF2 | 49.8s | 20 | 16.6min | 19.61s |

## F.2 SYNTHETIC DATA EXPERIMENTS

Table 4: MMD for observational, interventional, and counterfactual time series forecasting across causal structures: Tree, Diamond, FC-Layer, and Chain. Reported values are averaged over 50 test batches, each containing 128 test series.

| | Tree | | | | Diamond | | | | FC-Layer | | | | Chain | | | |
|---|---|---|---|---|---|---|---|---|---|---|---|---|---|---|---|---|
| | Additive | | NLNA | | Additive | | NLNA | | Additive | | NLNA | | Additive | | NLNA | |
| | Obs. | Int. | Obs. | Int. | Obs. | Int. | Obs. | Int. | Obs. | Int. | Obs. | Int. | Obs. | Int. | Obs. | Int. |
| DoFlow | $0.07_{\pm.01}$ | $\mathbf{0.07_{\pm.01}}$ | $\mathbf{0.11_{\pm.03}}$ | $\mathbf{0.13_{\pm.03}}$ | $0.03_{\pm.01}$ | $0.05_{\pm.01}$ | $\mathbf{0.14_{\pm.01}}$ | $\mathbf{0.17_{\pm.04}}$ | $\mathbf{0.01_{\pm.00}}$ | $\mathbf{0.03_{\pm.01}}$ | $0.14_{\pm.02}$ | $\mathbf{0.12_{\pm.02}}$ | $\mathbf{0.09_{\pm.02}}$ | $\mathbf{0.11_{\pm.03}}$ | $\mathbf{0.17_{\pm.03}}$ | $\mathbf{0.19_{\pm.03}}$ |
| GRU | $0.12_{\pm.02}$ | $0.14_{\pm.03}$ | $0.19_{\pm.04}$ | $0.25_{\pm.07}$ | $0.05_{\pm.01}$ | $0.10_{\pm.03}$ | $0.19_{\pm.03}$ | $0.24_{\pm.07}$ | $\mathbf{0.01_{\pm.00}}$ | $0.16_{\pm.04}$ | $0.19_{\pm.04}$ | $0.27_{\pm.10}$ | $\mathbf{0.10_{\pm.02}}$ | $0.15_{\pm.06}$ | $0.25_{\pm.06}$ | $0.31_{\pm.13}$ |
| TFT | $\mathbf{0.08_{\pm.01}}$ | $0.13_{\pm.04}$ | $0.18_{\pm.05}$ | $0.25_{\pm.09}$ | $0.08_{\pm.03}$ | $0.15_{\pm.05}$ | $0.20_{\pm.05}$ | $0.26_{\pm.10}$ | $0.06_{\pm.02}$ | $0.18_{\pm.06}$ | $0.20_{\pm.06}$ | $0.29_{\pm.13}$ | $0.13_{\pm.03}$ | $0.19_{\pm.09}$ | $0.23_{\pm.05}$ | $0.30_{\pm.12}$ |
| TiDE | $0.09_{\pm.02}$ | $0.14_{\pm.04}$ | $0.19_{\pm.05}$ | $0.24_{\pm.09}$ | $\mathbf{0.02_{\pm.01}}$ | $0.12_{\pm.04}$ | $0.16_{\pm.05}$ | $0.27_{\pm.13}$ | $0.03_{\pm.01}$ | $0.16_{\pm.05}$ | $0.22_{\pm.06}$ | $0.30_{\pm.12}$ | $0.13_{\pm.02}$ | $0.18_{\pm.07}$ | $0.22_{\pm.05}$ | $0.32_{\pm.10}$ |
| TSMixer | $0.10_{\pm.03}$ | $0.13_{\pm.03}$ | $0.18_{\pm.04}$ | $0.24_{\pm.08}$ | $\mathbf{0.02_{\pm.01}}$ | $0.14_{\pm.05}$ | $0.17_{\pm.05}$ | $0.26_{\pm.11}$ | $0.03_{\pm.01}$ | $0.18_{\pm.05}$ | $0.19_{\pm.04}$ | $0.32_{\pm.10}$ | $0.15_{\pm.03}$ | $0.20_{\pm.06}$ | $0.26_{\pm.06}$ | $0.35_{\pm.10}$ |
| DeepVAR | $0.12_{\pm.02}$ | $0.10_{\pm.02}$ | $0.18_{\pm.03}$ | $0.19_{\pm.03}$ | $0.10_{\pm.02}$ | $0.11_{\pm.02}$ | $0.22_{\pm.04}$ | $0.25_{\pm.08}$ | $0.08_{\pm.02}$ | $0.23_{\pm.04}$ | $0.26_{\pm.06}$ | $0.35_{\pm.10}$ | $0.11_{\pm.02}$ | $0.16_{\pm.03}$ | $0.24_{\pm.05}$ | $0.29_{\pm.05}$ |
| MQF2 | $\mathbf{0.08_{\pm.01}}$ | $0.16_{\pm.03}$ | $0.20_{\pm.03}$ | $0.29_{\pm.06}$ | $0.09_{\pm.02}$ | $0.15_{\pm.04}$ | $0.19_{\pm.03}$ | $0.30_{\pm.10}$ | $0.07_{\pm.01}$ | $0.20_{\pm.03}$ | $0.21_{\pm.03}$ | $0.34_{\pm.09}$ | $0.17_{\pm.03}$ | $0.22_{\pm.05}$ | $0.28_{\pm.04}$ | $0.33_{\pm.07}$ |

Table 5: CRPS for observational and interventional time–series forecasting across causal structures: Tree, Diamond, FC-Layer, and Chain. Reported values are averaged over 50 test batches, each containing 128 test series.

| | Tree | | | | Diamond | | | | FC-Layer | | | | Chain | | | |
|---|---|---|---|---|---|---|---|---|---|---|---|---|---|---|---|---|
| | Additive | | NLNA | | Additive | | NLNA | | Additive | | NLNA | | Additive | | NLNA | |
| | Obs. | Int. | Obs. | Int. | Obs. | Int. | Obs. | Int. | Obs. | Int. | Obs. | Int. | Obs. | Int. | Obs. | Int. |
| DoFlow | $0.22_{\pm.01}$ | $\mathbf{0.21_{\pm.01}}$ | $\mathbf{0.27_{\pm.02}}$ | $\mathbf{0.29_{\pm.02}}$ | $\mathbf{0.13_{\pm.01}}$ | $\mathbf{0.15_{\pm.01}}$ | $\mathbf{0.17_{\pm.02}}$ | $\mathbf{0.20_{\pm.02}}$ | $\mathbf{0.11_{\pm.01}}$ | $\mathbf{0.14_{\pm.01}}$ | $\mathbf{0.16_{\pm.02}}$ | $\mathbf{0.14_{\pm.02}}$ | $\mathbf{0.24_{\pm.02}}$ | $\mathbf{0.25_{\pm.02}}$ | $\mathbf{0.38_{\pm.04}}$ | $\mathbf{0.39_{\pm.04}}$ |
| DeepVAR | $0.24_{\pm.02}$ | $0.27_{\pm.02}$ | $0.32_{\pm.03}$ | $0.35_{\pm.03}$ | $0.16_{\pm.01}$ | $0.24_{\pm.02}$ | $0.21_{\pm.02}$ | $0.26_{\pm.03}$ | $0.14_{\pm.01}$ | $0.19_{\pm.02}$ | $0.20_{\pm.02}$ | $0.25_{\pm.03}$ | $0.27_{\pm.02}$ | $0.29_{\pm.02}$ | $0.44_{\pm.05}$ | $0.47_{\pm.06}$ |
| MQF2 | $\mathbf{0.19_{\pm.02}}$ | $0.29_{\pm.02}$ | $0.34_{\pm.03}$ | $0.40_{\pm.04}$ | $0.17_{\pm.02}$ | $0.26_{\pm.03}$ | $0.23_{\pm.02}$ | $0.31_{\pm.04}$ | $0.13_{\pm.01}$ | $0.21_{\pm.02}$ | $0.22_{\pm.02}$ | $0.29_{\pm.03}$ | $0.28_{\pm.03}$ | $0.32_{\pm.03}$ | $0.46_{\pm.05}$ | $0.52_{\pm.07}$ |

Table 6: RMSE for observational, interventional, and counterfactual time-series forecasting on the Chain causal structure.

| | Chain | | | | | |
|---|---|---|---|---|---|---|
| | Additive | | | NLNA | | |
| | Obs. | Int. | CF. | Obs. | Int. | CF. |
| **DoFlow** | $\mathbf{0.61_{\pm.13}}$ | $\mathbf{0.66_{\pm.13}}$ | $\mathbf{0.30_{\pm.06}}$ | $\mathbf{0.69_{\pm.16}}$ | $\mathbf{0.71_{\pm.17}}$ | $\mathbf{0.41_{\pm.07}}$ |
| GRU | $0.68_{\pm.11}$ | $1.01_{\pm.14}$ | NA | $0.80_{\pm.11}$ | $1.21_{\pm.16}$ | NA |
| TFT | $0.63_{\pm.17}$ | $1.10_{\pm.24}$ | NA | $0.78_{\pm.15}$ | $1.29_{\pm.19}$ | NA |
| TiDE | $0.65_{\pm.10}$ | $1.07_{\pm.17}$ | NA | $0.77_{\pm.12}$ | $1.16_{\pm.18}$ | NA |
| TSMixer | $0.67_{\pm.11}$ | $1.09_{\pm.18}$ | NA | $0.75_{\pm.14}$ | $1.20_{\pm.20}$ | NA |
| DeepVAR | $\mathbf{0.62_{\pm.12}}$ | $0.97_{\pm.18}$ | NA | $0.84_{\pm.13}$ | $1.09_{\pm.16}$ | NA |
| MQF2 | $0.73_{\pm.13}$ | $1.18_{\pm.19}$ | NA | $0.90_{\pm.15}$ | $1.30_{\pm.21}$ | NA |

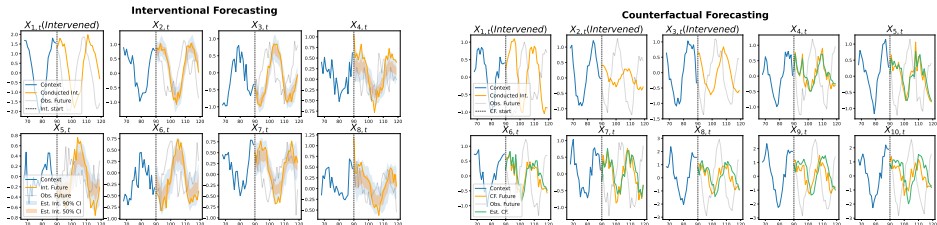

Figure 8: *Left:* "**Tree**" interventional forecasting results. Node $X_{1,t}$ is intervened. DoFlow provides 50% and 90% prediction intervals; the orange lines indicate the true interventional future. *Right:* "**Layer**" counterfactual forecasting results. Nodes $X_{1,t}$, $X_{2,t}$, and $X_{3,t}$ are intervened. DoFlow provides a single forecast in green; the orange lines indicate the true counterfactual future.

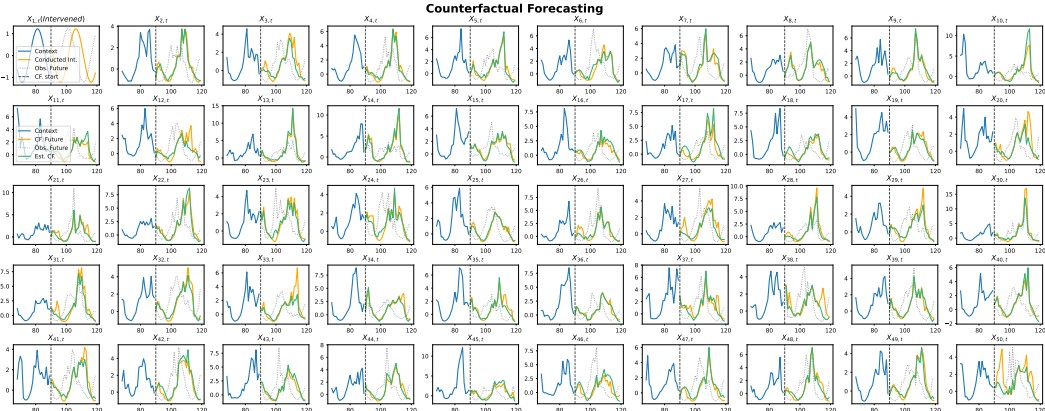

Figure 9: "**Chain**" (NLNA) counterfactual forecasting results. Node $X_{1,t}$ is intervened. DoFlow provides a single forecast in green; the orange lines indicate the true counterfactual future.

### F.3 HYDROPOWER SYSTEM

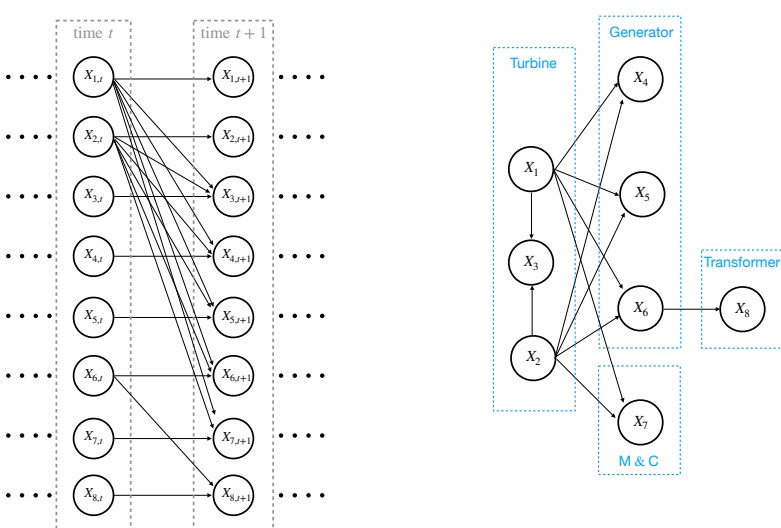

Figure 10: **Hydropower** system graph over 8 nodes. Exogenous variables $U_{i,t}$ are omitted for clarity but exist for every node at each time $t$. **Left:** Full node-level causal structure between consecutive time, with all variables $\{X_{1,t}, \ldots, X_{8,t}\}$ present at each step. **Right:** Rolled-up (time-suppressed) view over different nodes $\{X_1, \ldots, X_8\}$. Each arrow $X_i \rightarrow X_j$ (with $i \neq j$) denotes a lag-1 temporal dependency $X_{i,t-1} \rightarrow X_{j,t}$ that holds for all $t$. Both panels depict the same underlying structure.

Reliable modeling and uncertainty quantification are increasingly central to modern signal processing (Dong et al., 2020; Nemani et al., 2023; Gao et al., 2023; Wu et al., 2024a). Here, we highlight a new perspective enabled by DoFlow, causal reasoning in time series, through a hydropower signal processing case study.

In Figure 10, $X_1$ and $X_2$ denote the horizontal and vertical rotational vibrations of the hydraulic turbine, which directly drive the generator's operation. $X_3$ represents an auxiliary turbine state that is affected by $X_1$ and $X_2$. The generator's horizontal and vertical dynamics are represented by $X_4$ and $X_5$, while $X_6$ captures the generator's current output delivered to the transformer ($X_8$) for voltage regulation and transmission to the power grid. The Metering and Control (M&C) unit ($X_7$) continuously monitors the turbine's performance and stability to ensure coordinated operation.

Table 7 reports the RMSE results for observational and interventional time-series forecasting in the hydropower system. For interventional forecasting, we use 12 real power outages where the root nodes $X_1$ and $X_2$ fail, causing the entire system to an outage. In this setting, the root nodes are treated as intervened by the breakdown signals. The reported averages and standard deviations are computed over 12 runs with batch size $B = 1$ for the interventional case, in contrast to 50 runs with batch size $B = 128$ for the observational case.

Notably, the hydropower signals are highly unstable, with turbine flow and generator readings often exhibiting abrupt jumps or burnouts without clear patterns. Consequently, all methods face difficulty in accurate prediction, and the relatively high RMSE values reflect this inherent challenge. Nevertheless, our model performs consistently better than others under these conditions.

|  | Hydropower System | |
|---|---|---|
|  | Obs. | Int. |
| **DoFlow** | **$1.13_{\pm.18}$** | **$1.21_{\pm.19}$** |
| GRU | $2.05_{\pm.32}$ | $2.45_{\pm.35}$ |
| TFT | $1.82_{\pm.25}$ | $2.16_{\pm.41}$ |
| TiDE | $1.49_{\pm.24}$ | $2.08_{\pm.40}$ |
| TSMixer | $1.51_{\pm.25}$ | $2.11_{\pm.32}$ |
| DeepVAR | $1.78_{\pm.26}$ | $2.39_{\pm.28}$ |
| MQF2 | $1.97_{\pm.24}$ | $2.62_{\pm.34}$ |

Table 7: RMSE for observational and interventional time-series forecasting in the hydropower system.

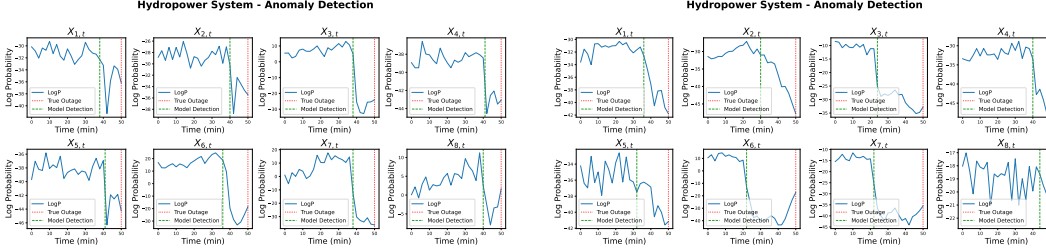

Figure 11: Anomaly detection by DoFlow on real power outages in the hydropower system (two segments shown).

For anomaly detection, we use DoFlow as a likelihood-based scoring model over *predicted future windows*. Fix a node $i$ and a current time $t$ in the test sequence. We treat the observed past $\{\mathbf{X}_1, \ldots, \mathbf{X}_t\}$ as context, sample a future window of length $L = 30$ minutes from DoFlow, $\{\hat{X}_{i,t+1}, \ldots, \hat{X}_{i,t+L}\}$, and compute the conditional log-density that DoFlow assigns to this *generated* forecast trajectory (i.e., the model-implied confidence of its own prediction) using the change-of-variables formula. Formally, this yields the node-wise anomaly score

$$\ell_i(t) := \log p_{\theta, X_{i,t+1:t+L}}\left(\hat{x}_{i,t+1:t+L} \mid \hat{H}_{i,t}\right),$$

where $\hat{x}_{i,t+1:t+L}$ denotes the forecast window decoded from sampled base noise $z_{i,t+1:t+L}$ under DoFlow conditioned on $\hat{H}_{i,t}$. Here, the log-likelihood is computed using the Eq. (13), restricted to

the window $[t+1, t+L]$:

$$\log p_{\theta, X_{i,t+1:t+L}}\left(\hat{x}_{i,t+1:t+L} \mid \hat{H}_{i,t}\right) = \sum_{u=t+1}^{t+L}\left[\log q(z_{i,u}) + \int_0^1 \nabla \cdot v_{i,\theta}\left(x_{i,u}(s), s; \hat{H}_{i,u-1}\right) ds\right], \quad (59)$$

with $\hat{x}_{i,u} = \Phi_\theta^{-1}(z_{i,u}; \hat{H}_{i,u-1})$ and $\hat{H}_{i,u-1}$ the RNN summary of the past up to $u-1$ as defined in Section 2.3. Intuitively, $\ell_i(t)$ measures how typical the entire upcoming 30-minute trajectory of node $i$ appears under the dynamics learned from normal-operation data by the model.

To obtain a quantitative decision rule, we first compute $\ell_i(t)$ on all test windows that do *not* overlap any outages and collect the empirical distribution of these scores for each node $i$. Let $\eta_i$ denote the 10th percentile of this normal-score distribution. We then flag time $t$ as *locally abnormal* for node $i$ whenever

$$\ell_i(t) < \eta_i.$$

To avoid reacting to isolated noise spikes, we declare an anomaly only if at least one node exhibits 5 consecutive minutes of low scores, i.e., there exists $i$ and a run of times $t, t+1, \ldots, t+4$ such that $\ell_i(\tau) < \eta_i$ for all $\tau$ in that run. The anomaly *detection time* for a given outage is the earliest $t$ at which this criterion is met; in Figure 11, the corresponding detection markers are plotted relative to the known outage time (minute 50 in each panel).

In Figure 11, notably, DoFlow's log-probability output becomes abnormal well before the outage occurs—sometimes as early as 20 minutes prior (e.g., $X_{6,t}$ in the right panel) and as late as 10 minutes prior. This allows anomalies to be detected in advance of the true outage.

### F.4 CANCER TREATMENT EFFECTS

We apply DoFlow to interventional forecasting of cancer tumor outcomes. For each test patient, the model observes the first 55 days of factual history $\{(\mathbf{X}_t, Y_t)\}_{t=1}^{55}$, where $\mathbf{X}_t = \{X_{i,t}\}_{i=1}^4$ represents chemotherapy and radiotherapy assignments and dosages, and $Y_t$ denotes the tumor volume.

Notably, unlike our other experiments, here only the tumor-volume node $Y_t$ is modeled with a CNF. The treatment actions $\mathbf{X}_t$ (chemotherapy and radiotherapy assignments/dosages) are fully observed and directly provided at every time step. We still use RNNs to summarize the joint history of past treatments $\mathbf{X}_t$ and past tumor outcomes $Y_t$ into a hidden state $\hat{H}_{t-1}$, which serves as the conditioning variable for the CNF of $Y_t$.

During the forecasting window (days 56–62), the treatment schedule is replaced by one of ten pre-defined intervention plans $\mathcal{I}^j = \{(i,t) : X_{i,t} \leftarrow m_{i,t}^j\}$, where $j$ indicates the $j$-th treatment plan and $j \in \{1, 2, \ldots, 10\}$. At each time step $t$, DoFlow estimates the interventional tumor outcome by sampling from the learned flow model:

$$\hat{Y}_t = \Phi_\theta^{-1}(z_t; \hat{H}_{t-1}), \qquad z_t \sim \mathcal{N}(0, I), \quad (60)$$

where $\Phi_\theta^{-1}$ denotes the learned reverse flow conditioned on the recurrent hidden state $\hat{H}_{t-1}$, which encodes the patient's historical outcomes and past treatments. The hidden state is updated autoregressively using the newly generated $\hat{Y}_t$ and the active treatments $\mathbf{X}_t = \mathbf{m_t^j}$.

The model implementations are detailed in Appendix E.4.

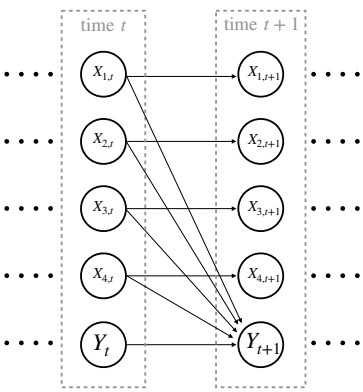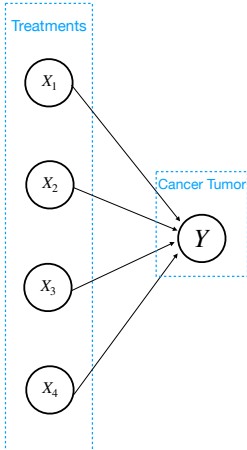

Figure 12: **Cancer Treatment** DAG over 8 nodes. Exogenous variables $U_{i,t}$ are omitted for clarity but exist for every node at each time $t$. **Left:** Full node-level causal structure between consecutive time, with all treatment variables $\{X_{i,t}\}_{i=1}^4$ and cancer tumor outcome $Y_t$ present at each step. **Right:** Rolled-up (time-suppressed) view over different nodes. Each arrow $X_i \rightarrow Y$ denotes a lag-1 temporal dependency $X_{i,t-1} \rightarrow Y_t$ that holds for all $t$. Both panels depict the same underlying structure.

Table 8: Normalized RMSE$_\tau$ for causal treatment effects on cancer tumor outcome. At future step $\tau$, RMSE is computed across all patient–option pairs as. Column groups represent the chemotherapy and radiotherapy application budgets $(\gamma_c, \gamma_r)$ in various data simulation scenarios.

| $\tau$ | $\gamma_c = 5$, $\gamma_r = 5$ | | | | $\gamma_c = 5$, $\gamma_r = 0$ | | | | $\gamma_c = 0$, $\gamma_r = 5$ | | | |
| --- | --- | --- | --- | --- | --- | --- | --- | --- | --- | --- | --- | --- |
| | DoFlow | CRN | RMSN | MSM | DoFlow | CRN | RMSN | MSM | DoFlow | CRN | RMSN | MSM |
| 3 | **1.25**% | 2.43% | 3.16% | 6.75% | **0.49**% | 1.08% | 1.35% | 3.68% | **0.94**% | 1.54% | 1.59% | 3.23% |
| 4 | **1.73**% | 2.83% | 3.95% | 7.65% | **0.76**% | 1.21% | 1.81% | 3.84% | **1.10**% | 1.81% | 2.25% | 3.52% |
| 5 | **2.08**% | 3.18% | 4.37% | 7.95% | **0.88**% | 1.33% | 2.13% | 3.91% | **1.27**% | 2.03% | 2.71% | 3.63% |
| 6 | **2.74**% | 3.51% | 5.61% | 8.19% | **1.09**% | 1.42% | 2.41% | 3.97% | **1.69**% | 2.23% | 2.73% | 3.71% |
| 7 | **3.22**% | 3.93% | 6.21% | 8.52% | **1.33**% | 1.53% | 2.43% | 4.04% | **2.01**% | 2.43% | 2.88% | 3.79% |

We compute the normalized root mean-squared error (RMSE) at the $\tau$-th step across all patients and treatment options as

$$\text{RMSE}_\tau = \frac{\sqrt{\frac{1}{N}\sum_{n=1}^N \left(Y_{n,t+\tau} - \hat{Y}_{n,t+\tau}\right)^2}}{\frac{1}{N}\sum_{n=1}^N Y_{n,t+\tau}}, \tag{61}$$

where $N$ is the total number of patient–option pairs in the test set. Smaller NRMSE indicates more accurate estimation of causal treatment effects.

Table 8 reports the normalized RMSE results for causal treatment effect estimation on cancer tumor outcomes. We compare our method with three established baselines: CRN (Counterfactual Recurrent Network) Bica et al. (2020a), RMSN (Recurrent Marginal Structural Network) Lim (2018), and MSM (Marginal Structural Model) Mansournia et al. (2012). The datasets follow the construction in Bica et al. (2020a), where $\gamma_c$ and $\gamma_r$ denote the treatment-application budgets for chemotherapy and radiotherapy, respectively. Some baseline results in Table 8 are adopted directly from Bica et al. (2020a).

