# OpenReview forum: "DoFlow: Flow-based Generative Models for Interventional and Counterfactual Forecasting on Time Series"
_ICLR.cc/2026/Conference — ICLR 2026 Poster_

### Official Review · Reviewer_Pjb9 · 2025-10-16

**Soundness:** 2
**Presentation:** 2
**Contribution:** 3
**Rating:** 4
**Confidence:** 2

**Summary:**

This paper proposes a novel architecture based on Normalizing Flows for time-series forecasting. The model is designed to handle three types of forecasting queries: observational (standard prediction), interventional (predicting the effect of an action), and counterfactual (predicting what would have happened under a different action, given a specific observed outcome).
While the idea is interesting and the authors provide an appropriate theoretical analysis of the method, I have some concerns about specifically the presentation, the empirical evaluation, and the motivation for counterfactual simulations that I would like to be addressed before recommending the paper for acceptance.

**Strengths:**

1.  **Novelty:** The core idea of using a unified Normalizing Flow-based framework to tackle observational, interventional, and counterfactual time-series forecasting is interesting and novel.
2.  **Theoretical Grounding:** The paper provides a theoretical analysis of the proposed method, which adds rigor to the contribution.

**Weaknesses:**

1. **Motivation and Practical Utility of Counterfactual Forecasting:** My most fundamental concern is the practical relevance of the counterfactual simulations as defined in the paper. The method hinges on conditioning on the exogenous noise `z` from an observed trajectory to ask "what if?" questions. However, the motivation for why a practitioner or decision-maker would need to perform a simulation conditioned on this specific noise instance is unclear.
    - Could the authors provide a concrete, practical decision-making scenario where this type of counterfactual simulation provides unique, actionable information that cannot be obtained from simpler interventional forecasts?
    - How does this concept translate to models that are not probabilistic (e.g., deterministic models)? Is intervening on a variable `X` sufficient for counterfactual reasoning if there are no further noise components?


2. **Insufficient Experimental Details and Reproducibility:** The empirical evaluation is missing critical information, making it difficult to assess the validity of the results.
    - **Hyperparameters and Model Specifications:** There are no details on the hyperparameters for the proposed model or for any of the baselines. Information on architecture, training duration, learning rates, etc., is essential. E.g. 908-910 is extremely vague and should be further specified.
    - **Dataset Details:** The paper lacks information on the amount of data used for training, validation, and testing for each experiment, nor does it provide information on whether such a split was actually used. For the real-world power outage data, it is unclear how the model was trained, as there is presumably only one historical trajectory. Was the power outage event included in the training data, or was the time series split? This needs further clarification.
    - **Intervention Details:** The nature of the interventions is not specified. Were the interventions in-distribution or out-of-distribution? How many variables were affected simultaneously? This context is crucial, especially given that the baselines appear to perform strikingly worse than the proposed method, even when trained on only the correct Parent set. A detailed description of the experimental setup is needed to understand this performance gap.

3. **Unclear Explanation of Anomaly Detection Results:** The application to preemptive anomaly detection is highlighted as a key result, but is not sufficiently explained.
    - How exactly is the detection performed? What quantity is measured, and what threshold defines an "abnormal" reading?
    - The claim of *preemptive* detection is extraordinary. Why can the model detect an anomaly before it occurs? Is this consistent with the physical reality of the system, or is it an artifact of the experimental setup (e.g., information leakage)? A more detailed explanation of the underlying mechanism is necessary to properly assess this result.

4. **Clarity and Connection of Paper Sections:** Certain parts of the paper seem disconnected or lack support.
    - **L60 (Causal Discovery):** The discussion of the causal discovery literature feels vague and disconnected from the paper's core contribution, as the method fundamentally assumes a known and fixed causal graph (DAG). The authors should either better integrate this discussion or remove it.
    - **Section 4.4:** This section appears unsupported and feels out of place, as it is not explained how exactly the anomaly detection can be performed.
    - **Methodological Limitations:** It should be explicitly noted and discussed that the proposed model does not account for instantaneous causal effects within a single time step, which is a significant assumption in many real-world systems.
    - **Table 1:** The table is overflowing and quite small. Please consider updating the formatting

**Questions:**

- As noted in the specified weaknesses.

- What happens if the causal graph is not fully known or partly wrongly specified? Is DoFlow able to recover from such misspecifications?

- How would you position your method against works such as [1] ?

- Could you imagine under which conditions (or data distributions) DoFlow is struggling? A discussion on this could help the reader assess the applicability of your method to their problem.

¹ https://arxiv.org/abs/2402.09891

---

> ### Author Response · Authors · 2025-11-17
> **Rely to Reviewer Pjb9**
>
> Thank you sincerely for your time and thoughtful review! Below, we respond to each of your questions in detail. The updated PDF also reflects our revisions to all of your concerns. We would greatly appreciate it if you could have a look!
>
> > Could the authors provide a concrete, practical decision-making scenario where this type of counterfactual simulation provides unique, actionable information that cannot be obtained from simpler interventional forecasts?
>
> A standard forecaster learns $p(X_{\tau+1:T} | x_{1:\tau})$, and once $x_{1:\tau}$ is fixed, produces the same forecast regardless of hypothetical changes, so it cannot answer counterfactual queries. We now have clearly explained interventional and counterfactual with detailed examples in **Lines 38-53**. Here we use **counterfactual** as a concrete illustration:
>
> Consider a healthcare setting where we observe a patient’s treatment history and outcome trajectory, and then ask whether this particular patient’s outcomes would have been better or worse under a different dosing schedule. Notably, outcomes depend not only on dosing but also on patient-specific factors that are **not directly observed** (e.g., baseline health), yet **these are reflected in the factual trajectory**. By conditioning on the observed factual trajectory, we can infer these unobserved factors, and then re-simulate how the same patient’s trajectory would have evolved under the alternative dosing plan.
>
> In DoFlow, we use the forward process (Eq. (11)) to encode $z_{i,t}^{F}$ from the factual trajectory, which captures the individual-specific **unobserved factors** inferred from the factual path. Then, using the reverse process (Eq. (12)) starting from $z_{i,t}^{F}$, we generate the individual-specific counterfactual trajectory under another intervention.
>
> Section 4 provides theoretical support for this counterfactual recovery. To our knowledge, DoFlow is the first to enable such counterfactual reasoning in time-series forecasting.
>
> > Is intervening on a variable $X$ sufficient for counterfactual reasoning if there are no further noise components?
>
> In an *idealized SCM* with fully known structural equations and no unobserved influences, intervening on $X$ alone is sufficient.
>
> However, in real-world systems it is almost never possible to specify a perfect SCM, i.e., to identify all true causes $X_{\text{pa}(i)}$ of each variable $X_{i}$. For this reason, we model each node as: $X_{i,t} := f_i(X_{i,t-}, X_{\mathrm{pa}(i),t-}, U_{i,t})$, where $U_{i,t}$ represents unobserved exogenous factors.
>
> Notably, our DoFlow is trained entirely on observational data, with only the parent–child structure provided. Despite never observing $U_{i,t}$, Proposition 4.3  shows that DoFlow learns these exogenous variables: through encoding the factual trajectory, it recovers a latent representation $Z_{i,t}$ that is in bijection with $U_{i,t}$.
>
> This is what modern observational forecasters (both deterministic and probabilistic ones) cannot do.
>
> > Thanks for your careful attention to our experimental details! Please see the information below on where these details can be found in our revised paper:
>
> > Hyperparameters and Model Specifications:
>
> These are now provided in the newly added "Appendix E: Model and Implementation Details."
>
> > Dataset Details and Intervention Details:
>
> The choices of $\beta$ used in generating the synthetic datasets are now detailed in the final paragraphs of Appendices D.1–D.4. The procedure for conducting interventions in the simulations is provided in Appendix D.5.
>
> Information on training and evaluation is now included in Appendix E. For details specific to anomaly detection, please refer to our responses to Weakness 3 below.
>
> We also provide the datasets associated with our code in the supplementary materials.
>
> > Unclear Explanation of Anomaly Detection Results:
>
> We have added a detailed explanation in Appendix F.4 (**Lines 1579–1607**). Briefly, DoFlow is trained exclusively on normal-operation data. By Proposition 3.1, at each time $t$, the model can compute the log-probability of the next 30-minute window under the learned normal dynamics. When an abnormal event occurs, the observed trajectory deviates from the normal patterns learned by the model, resulting in a significant drop in log-probability output.
>
> > Clarity on Causal Discovery literature:
>
> In the initial version of the paper, we categorized related causal time–series work into three groups: (1) causal effects, (2) counterfactual explainability, and (3) causal discovery. As also suggested by Reviewer jniY, we have removed Category 2 in the revised version to avoid distracting readers.
>
> We chose to keep *Causal discovery* literature in the discussion, since if the causal DAG is unknown, readers may use *causal discovery* methods to construct a temporal causal graph from observational data before applying our method. We now explicitly discuss this connection in *Limitations* of Section 5.4.

---

> ### Author Response · Authors · 2025-11-17
> **Rely to Reviewer Pjb9 (Second Part)**
>
> > Please see our continued responses to your concerns. Thanks for your time!
>
> > Section 4.4 (now Section 3.3) on Anomaly Detection:
>
> We fully agree with your concerns. We have renamed Section 3.3 to “Additional Property: Likelihood-Based Anomaly Detection” for clarity. As you suggested, we also added a detailed explanation of how the anomaly detection procedure is performed in Appendix F.4.
>
> > Methodological Limitations: It should be explicitly noted and discussed that the proposed model does not account for instantaneous causal effects within a single time step.
>
> We have now added this assumption in Lines 136-137.
>
> > What happens if the causal graph is not fully known or partly wrongly specified? Is DoFlow able to recover from such misspecifications?
>
> As stated in Section 1 and Section 2.1, DoFlow assumes a known causal DAG; the current paper does not address settings where the graph is unknown or misspecified.
>
> To our knowledge, no prior method has modeled the counterfactual time-series trajectory defined in Eq. (4). We believe DoFlow represents a meaningful step forward by bringing interventional and counterfactual forecasting into time-series modeling.
>
> We now explicitly list this as a *Limitation* in Section 5.4. When the causal structure is unknown, existing causal discovery methods can serve as an upstream task to construct temporal causal DAG, before applying DoFlow. As noted in Section 5.4, an meaningful future direction is to integrate DoFlow with deconfounding causal generative models so that the model can learn the complete temporal structures during causal inference, for more reliable interventional and counterfactual reasoning.
>
> > How would you position your method against works such as [1] ?
>
> The work “Do causal predictors generalize better to new domains?” focuses on *static*, *non–time-series* settings, and investigates whether restricting models to causal features improves domain generalization accuracy.
>
> In contrast, DoFlow addresses a different problem: it is one of the first methods to bring interventional and counterfactual forecasting into *time-series*. Following the suggestions of Reviewer jniY and Reviewer dDVY, we have expanded Section 1.1 (Related Works) and clarified distinctions, explicitly noting what is missing in prior work (Lines 114–117) and more clearly outlining the contributions of our DoFlow.
>
> > Could you imagine under which conditions (or data distributions) DoFlow is struggling? A discussion on this could help the reader assess the applicability of your method to their problem.
>
> We now explicitly list the assumption of a known causal DAG as a limitation in Section 5.4 and outline corresponding future directions.
>
> In addition, Appendix B.1 (Lines 931-962) clarifies the standard SCM assumptions underlying DoFlow, especially the requirement of *unconfoundedness*: all common causes of the observed variables are included in the DAG. We further added Remark B.2 (Lines 949-962), which discusses this assumption and highlights that extending DoFlow beyond unconfoundedness assumption, to handle partially unknown structures, is an important direction for future work.
>
> > Below, we summarize our **key contributions**:
>
> 1. *Introducing interventional and counterfactual forecasting to time series*:
>
> We bring interventional and counterfactual reasoning into time-series forecasting—an area where interventional forecasting has been under-explored (existing works focus mainly on treatment-effect estimation), and where counterfactual forecasting, to our knowledge, has not been explored in prior work.
>
> 2. *Theoretical guarantees for counterfactual recovery*:
>
> We establish formal counterfactual recovery properties of DoFlow in Section 4 under Assumption 4.1, showing that the model can correctly identify exogenous noise and recover exact counterfactual trajectories.
>
> 3. *Comprehensive empirical evaluation on synthetic and real-world data*:
>
> We conduct extensive synthetic experiments with quantitative metrics, evaluate observational and interventional forecasting on a real Hydropower dataset. Besides, new in the revision, we demonstrate DoFlow’s applicability to causal-effects problems through **a new real-world experiment** on *cancer treatment outcomes* (Section 5.3).
>
> We sincerely appreciate your valuable time and feedback! We look forward to any further comments you may have!

---

> > ### Comment · Reviewer_Pjb9 · 2025-11-25
> >
> > I thank the authors for their thorough rebuttal and the valuable additional experiments they provided. Most of my concerns have been addressed adequately, and I have raised my score accordingly. While my point on baseline specifications remains, I think it is not critical given the novelty of the area.
> >
> > My final suggestion pertains to the challenge of learning counterfactual distributions.
> > My last remaining concern is about the general nature of learning counterfactual distributions.
> > While the paper presents some empirical results on this, I find it unintuitive to trust such learned distributions even for a known DAG, especially in real-world applications. With the Chausal hierarchy [1] in mind, and the fact that one has to rely on observational distributions to learn some parts of the counterfactual distribution, commenting on this could further enhance the manuscript, especially since counterfactual forecasting is a relatively novel field.
> >
> > [1] Bareinboim, Elias, et al. "On Pearl’s hierarchy and the foundations of causal inference." Probabilistic and causal inference: the works of judea pearl. 2022. 507-556.

---

> > > ### Author Response · Authors · 2025-11-26
> > >
> > > Thank you for this insightful follow-up suggestion! Indeed, counterfactual distributions are generally not identifiable from observational data alone, even with a known DAG, unless additional structural assumptions are imposed on the underlying SCM.
> > >
> > > This limitation is shared by all model-based counterfactual methods. Counterfactual methods on static data can only validate counterfactual performance empirically on various simulated DAGs, and theoretical identifiability results can only be obtained under specific structural assumptions about the true data-generating SCM.
> > >
> > > Similarly, DoFlow is the first to extend counterfactual to time-series and demonstrate strong empirical performance across multiple DAG structures. In Section 4, we establish counterfactual identifiability under the assumption of a strictly monotone and continuous SCM $f(\cdot, U_{t})$. Beyond this regime, learned counterfactuals become model-dependent rather than theoretically identifiable. Therefore, an important future direction is to broader theoretical identifiability results.
> > >
> > > We have now explicitly added this discussion in the manuscript (Lines 518–525). Thank you once again for your time and careful review!

---

### Official Review · Reviewer_jniY · 2025-10-30

**Soundness:** 3
**Presentation:** 2
**Contribution:** 3
**Rating:** 6
**Confidence:** 2

**Summary:**

This paper proposes _DoFlow_ a framework based in NeuralODEs and Recurrent Neural Networks (RNN), to estimate not only observational forecasting distributions, but also interventional distributions and counterfactuals, opening the door to reliable decision making in decision making with time-series.

DoFlow models each variable in a known causal graph with its own NeuralODE, and conditions generation on a recurrent state that summarizes the past of that variable and its parents. This lets the model similate future trajectories in topological order. This allows to forecast observational series, interventional distributions and individual counterfactual outcomes. Because the model is a continuous normalizing flow, it also assigns likelihoods to predicted trajectories, which they use for anomaly and change-point detection. The authors provide a recovery result guaranteeing correct counterfactual reconstruction under certain assumptions, and they report experiments on synthetic causal time series and a real hydropower system. DoFlow matches or improves standard forecasting error compared to strong baselines, and uniquely supports interventional simulation, counterfactual rollout for each individual sequence, and early anomaly warnings before system failures.

**Strengths:**

- Preliminaries about NeuralODE  and Flow matching are clear. Additionally, the method section: 'Time conditioned flow on a causal DAG'  is very well written and the method is easy to understand. The method is based on state of the art deep learning techniques, and all the proposed strategy is sound and practical. I find the model useful for future reseach. However, I would like to see a complete scheme of the implementation of the framework: how many NeuralODEs are needed, how the RNN is trained and combined with the Flow, etc.

- The paper presents theoretical guarantees about the recoverability of counterfactuals, which are sound and mostly (see weaknesses) clear, although I have not checked the proofs of the Appendix in detail.

- The results on synthetic data support the theoretical insights in interventional forecasting and in counterfactual estimation. Also, experiments on real-world data demonstrate an impresive performance, not only in estimating interventions, but also in estimating anomaly detection via likelihoods. Experiments are systematic, extensive and complex enough.

**Weaknesses:**

- Neither the problem or the solution proposed are well specified in the introduction. There are many unconected paragraphs that do not provide a connected story line about the motivation, the **limitations of existing work** or the strategy that this paper proposes. For example, `line 96`says: ''motivated by these gaps...'', which gaps? Also, more information about the interventions that the model is suposed to handle has to be provided. What does '...counterfactual queries across the entire system'(`line 97-98`) mean?

- I have a minor concern about the theory section, although I probably miss something. First of all, I do not see a condition of invertibility between X and U. Assumption A2 requires monotone $f(\cdot, U_t)$. However, that is invertible only if the function is continuous and also if X is continous. How can we recover the value of the exogenous variable if the function is not bijective? I think that bijectivity should be required explicitly.

- There are implicit assumptions that have not being discussed in the paper:
1) Unconfoundedness / causal sufficiency. I.e., that all confounders are observed. If there exist time-varying confounders that are not observed, I do not see how the model can recover interventional distributions and counterfactuals.
2) Positivity. If the observational data is not rich enough, that is, if the intervention is done in a point in which the density of the observational distribution is not zero, the model would extrapolate the causal effects.
3) no interference and consistency: the classical SUTVA assumptions.
4) Regularity conditions: I think that the functions of the SCM have to meet some regularity conditions in order this theory to hold.


> Summary

Although I have some concerns---specially with the introduction and the related work section---and many questions, I find the paper a good contribution to causal inference in time series. I think the employment of neuralODEs to causal time series forecasting, with the inclusion of DAGs, which allows for interventional and counterfactual inference, is great step forward in causal forecasting. For all the previous reasons, I will recommend to accept the paper, and I will be happy to raise my score if my concerns are covered.

**Questions:**

- I do not really understand the implication that _counterfactual explanations_ in this work. Although the term _counterfactual_ is a common term in causal literature, counterfactual explanations are non-causal concepts that look for perturbations in the input to get the desired output. Is it necessary to include that in the introduction? I think those papers only introduce noise to the reader.

- Related with the previous point, I have the same question with causal discovery literature. Although causal discovery is a much more related topic, why do you include that in the introduction, if you assume a known causal graph? I think the introduction is scarce, in general very noisy, and the objective of the work is not clear.

- I am surprised because in the related work the authors name some models that solve interventional and counterfactual queries, but they do not name two models that are the state-of-the-art in this field and are based in normalizing flows, thus being very connected with the technologies of this paper: [1, 2]. Is there any reason for not including them?

- In general, some related works are missing or they lack discussion. For example, haven't authors considered to include [3]? Or can authors discuss the limitations of [4]? Those are only some examples, but more discussion have to be added in the introduction/ related work.

- Should the paper of continuous normalizing flows not be cited in line 115?

- Although section 4 is well written in general, $\Phi_\theta$ should be defined before introducing them in the algorithms. Otherwise, the reading is more difficult. Instead, it is defined in section 5 (`line 322`).

- Assumption A3 requires $Z\sim N(0,1)$, but is that distribution necessary? Is not enough with having Z independent from $H_{t-1}$, independently of the base distribution of $Z$?

- Everything that the DAG is assumed to be true, there is a question that arises: what if the DAG is not well specified? I think that some discussion about this fact would help.

- Have the authors thought of exploring the violation of 'all confounders observed'? Some work on this have been explored in causal generative models: [5, 6]

> References

[1] Javaloy, A., Sánchez-Martín, P., & Valera, I. (2023). Causal normalizing flows: from theory to practice. Advances in Neural Information Processing Systems, 36, 58833-58864.

[2] Khemakhem, I., Monti, R., Leech, R., & Hyvarinen, A. (2021, March). Causal autoregressive flows. In International conference on artificial intelligence and statistics (pp. 3520-3528). PMLR.

[3] Liu, Y., Sun, Y., & Lim, J. H. (2023, June). Counterfactual Dynamics Forecasting–a New Setting of Quantitative Reasoning. In Proceedings of the AAAI Conference on Artificial Intelligence (Vol. 37, No. 2, pp. 1764-1771).


[4] Shenghao Wu, Wenbin Zhou, Minshuo Chen, and Shixiang Zhu. Counterfactual generative models
for time-varying treatments. In Proceedings of the 30th ACM SIGKDD Conference on Knowledge
Discovery and Data Mining, pages 3402–3413, 2024

[5] Almodóvar, A., Javaloy, A., Parras, J., Zazo, S., & Valera, I. (2025). DeCaFlow: A Deconfounding Causal Generative Model. arXiv preprint arXiv:2503.15114.

[6] Kevin Muyuan Xia, Yushu Pan, and Elias Bareinboim. Neural Causal Models for Counterfactual Identification and Estimation. In The Eleventh International Conference on Learning Representations, ICLR 2023

---

> ### Author Response · Authors · 2025-11-17
> **Reply to Reviewer jniY**
>
> We sincerely thank you for the valuable comments and suggestions! We have thoroughly revised the draft accordingly, particularly the Introduction and Related Works. We would be very grateful if you could take a look at it. Below, we summarize where the key changes appear in the revised draft and provide detailed responses to your questions:
>
> > Neither the problem nor the solution proposed are well specified in the introduction.
>
> We have thoroughly rewritten Section 1 (Introduction) and Section 1.1 (Related Works), addressing all of your suggestions. We now clearly define interventional and counterfactual queries and provide detailed examples to clarify their meaning (Lines 38–53). We then explain what is missing in modern forecasters (Lines 54–58) and explicitly summarize the contributions of our work (Lines 59–65).
>
> We have also reorganized Section 1.1 (Related Works). We now begin by introducing modern *Time-series forecasting* methods (Lines 72–82) and *Causal generative modeling* on static data (Lines 83–91). We then present two research lines complementary to ours: *Causal effects on time series* and *Causal discovery*. After that, we clearly articulate what is missing in current causal time-series literature (Lines 113–120).
>
> > I have a minor concern about the theory section. First of all, I do not see a condition of invertibility between X and U. Assumption A2 requires monotone $f(\cdot, U_t)$. However, that is invertible only if the function is continuous and also if X is continous.
>
> We now revise the (A2) in Assumption 4.1 to: $f(\cdot, U_t)$ is *strictly monotone* and *continuous* in $U_t$. Therefore, under (A2), for any fixed $(X_{t-}, X_{\mathrm{pa},t-})$, the map $u \mapsto f(X_{t-}, X_{\mathrm{pa},t-}, u)$ is a bijection on the support of $U_t$. Thank you for helping us clarify this point!
>
> > There are implicit assumptions that have not been discussed in the paper.
>
> You are absolutely right. DoFlow is fundamentally built on an SCM framework, which implicitly entails assumptions on unconfoundedness, consistency, positivity, and no interference. We now explicitly present these assumptions and discuss them in Appendix B.1 (Lines 931-962), with clear references in the main text (Lines 343–347).
>
> We also elaborate on the *unconfoundedness* assumption in Remark B.2 and outline how future work could integrate DoFlow with approaches such as DecaFlow or Neural Causal Models to handle deconfounded or partially latent temporal DAGs.
>
> > I do not really understand the implication that counterfactual explanations in this work.
>
> Thanks for your suggestion! We have now removed this line of literature, as it is not related to our work, to avoid introducing unnecessary noise for readers.
>
> > Related with the previous point, I have the same question with causal discovery literature.
>
> We chose to keep this line of work in the discussion, since if the causal DAG is unknown, readers may use *causal discovery* methods to construct a temporal causal graph from observational data before applying our method. We now explicitly discuss this connection in *Limitations* of Section 5.4.
>
> > I am surprised because in the related work the authors name some models that solve interventional and counterfactual queries, but they do not name two models that are the state-of-the-art in this field.
>
> We appreciate this observation. We have now added references to the two state-of-the-art models in the *Causal generative modeling* subsection of Section 1.1 (Related Works). Thank you for pointing out this oversight!
>
> > In general, some related works are missing or they lack discussion. For example, haven't authors considered to include [3]? Or can authors discuss the limitations of [4]?
>
> Thanks for the comments! Work [4], "Counterfactual generative models for time-varying treatments" falls under the causal effects on time series category, and we have now included and discussed it in the Section 1.1 Related Works.
>
> Work [3], "Counterfactual Dynamics Forecasting", studies counterfactual dynamics in simulated physical systems, modeling object locations $p(t)$ and velocity $v(t)$ using their continuous dynamics in Equation (1). Importantly, their method is *trained on paired factual/counterfactual trajectories*, rather than providing a general multivariate time-series generator on an arbitrary causal DAG.
>
> > Should the paper of continuous normalizing flows not be cited in line 115?
>
> We now have cited Neural ODE in Line 172 when first introducing CNF.
>
> > Although section 4 is well written in general, $\Phi_{\theta}$ should be defined before introducing them in the algorithms.
>
> Thank you for the observation. We have now defined them clearly in equation (11) of "Forward Process" and in equation (12) of "Reverse Process", beginning at Line 228.

---

> > ### Author Response · Authors · 2025-11-17
> > **Reply to Reviewer jniY (Second Part)**
> >
> > > Assumption A3 requires $Z\sim N(0,1)$, but is that distribution necessary?
> >
> > Thanks for your attention to these details! This is not necessary, and we have now simply assumed that $p_{\theta}(Z_t\mid H_{t-1})=q(Z_t)$ in (A3).
> >
> > > What if the DAG is not well specified? Some discussion about this fact would help. Have the authors explored the violation of unconfoundedness?
> >
> > Thank you for raising this important point. We now include a discussion of the *Limitations* in Section 5.4. We also added Appendix B.1 on the SCM assumptions, as well as Remark B.2 explaining what happens when the *unconfoundedness* condition fails and pointing to possible future work.
> >
> > > Thank you sincerely again for all of your thoughtful comments! We also want to briefly highlight how we revised other parts of the draft to address the concerns raised by other reviewers:
> >
> > 1. We have thoroughly rewritten the Introduction and Related Works sections to position our problem more clearly, distinguish our setting from existing causal time-series work, and better articulate our contributions. In addition to your comments, we incorporated feedback from Reviewer sw48, Reviewer dDVY, and Reviewer Pjb9 as well.
> >
> > 2. To ensure full reproducibility, we expanded Appendix D by specifying how the parameters for the synthetic datasets were chosen and how interventions are conducted (Appendix D.5). We also added a new section—Appendix E: Model and Implementation Details, to provide complete hyperparameter and NN settings. These are suggested by Reviewer dDVY and Reviewer Pjb9.
> >
> > 3. For a more comprehensive empirical evaluation, we added the CRPS metric to Table 6 and Appendix D.6.3, as suggested by Reviewer sw48. Furthermore, in the revised version, we demonstrate DoFlow’s applicability to causal-effects problems through a **new real-world experiment** on *cancer treatment outcomes* (Section 5.4).
> >
> > Thank you once again for your valuable time. We truly appreciate it!

---

> > > ### Comment · Reviewer_jniY · 2025-11-24
> > > **response acknowledgment**
> > >
> > > Dear authors,
> > > I acknowledge your response to my review. I think the main concerns that I had have been addressed: the introduction is much clearer and other minor concerns have been improved. I think the current version of the manuscript is a good contribution in general.

---

> > > > ### Author Response · Authors · 2025-11-26
> > > >
> > > > Dear Reviewer jniY,
> > > >
> > > > Thank you very much for reading the revised version and for the encouraging comments! We are very glad to hear that the clarified introduction and other changes have addressed your concerns. We also truly appreciate your recognition of the paper as a good contribution to causal inference in time series.
> > > >
> > > > If there is any remaining aspect that you feel could still be further clarified, we would be very happy to address it. Thank you again for your thoughtful review throughout the process!

---

### Official Review · Reviewer_dDVY · 2025-10-30

**Soundness:** 2
**Presentation:** 1
**Contribution:** 2
**Rating:** 2
**Confidence:** 5

**Summary:**

The objective of this work, as expressed in Equation (9), is to obtain the counterfactual forecast $X_{\tau+1:T}^{\mathrm{CF}}$, given the interventions $\mathrm{do}(X_I := \gamma_I)$ applied to the predictor $X_{1:\tau}$ and the corresponding factual forecast $X_{\tau+1:T}^{\mathrm{F}}$, along with optional conditional variabels.  The authors trained a mapping that connects the current state to its past states and parent variables, based on a directed acyclic graph (DAG), while optionally incorporating conditional variables.  Using these factors as inputs, the authors employed flow matching to generate the current state as a probability distribution over possible outcomes. This process is autoregressive. To support the training of the entire pipeline, the authors used simulated data generated from four causal patterns, encompassing both linear and nonlinear additive models. After the entire framework is trained, given the essential inputs and specified interventions,  it can generate the corresponding counterfactual forecast distribution.

**Strengths:**

1. This paper introduces an unexplored research direction, as claimed by the authors, which could be of potential value.

2. The technical pipeline presented in this work appears to be complete.

3. Producing the probability density function (PDF) instead of a single counterfactual forecast is an insightful approach.

**Weaknesses:**

1. Several key concepts are not clearly introduced at the beginning. For example, in the Introduction section, it is unclear what is meant by “interventional question.” How are “binary or discrete, fixed-time actions, treatment focus …” defined? What does “extrapolating a counterfactual path and contrasting it” mean? Most importantly, the key concept of “counterfactual” should be clearly illustrated from the beginning. What does “these works do not encode causal DAGs or simulate system-wide trajectories under interventions” mean?

2. In the Introduction, I would prefer the authors to directly explain why the research problem is important, rather than simply stating that it has not been studied and is important in many fields without sufficient evidence or support. The authors are also advised to briefly outline the key challenges of the problem and how this paper addresses them, instead of letting the related work take up a large portion of the Introduction.

3. Why are the contributions placed in the Related Work section instead of the Introduction? Why does the Related Work section repeat content that has already been mentioned in the Introduction? In addition, the authors have not conducted a sufficiently thorough review of related work. Does the proposed method fall under an existing category of related work? How does it differ from similar studies?

4. It is recommended that the authors introduce the concept of a causal DAG in the Preliminary section. It should explain more on how a multivariate time series can be represented as a directed graph. It sounds like paths in a graph, but can we still name them as time series?

5. For the technical representation, the authors did not explain the meaning of the notation $\gamma$ in equation (8), and did not explain the difference between notations X and x. The meaning of h should be explained early for equation (10), where it first appeared. Equation (11) is not clear. Is it a concatenation of vectors of scales or a mapping? What is the specific use of the conditions? In equation (11), I suggest the authors to remind the readers the meaning of s clearly: the time step of flow matching or the time series. The authors also did not explain the meaning of K nodes. How do we construct the nodes? What are they representing?

**Questions:**

1. Please address the presentation issues mentioned above.

2. According to the mechanism of generative AI, particularly the flow-matching approach adopted in this work, the output represents only a sample from a potential distribution. Would this introduce additional uncertainty that complicates interpretability? Even if the analysis relies solely on the probability density function (PDF) to quantify significance, there remains a potential limitation regarding practical applicability: during training, small perturbations in past states may lead to extreme events in the future, and vice versa. Since such cases are statistical outliers, how can the authors justify that the trained flow-matching model is capable of capturing these rare patterns? If not, there is a risk that the forecasted extremes may be erroneously dismissed when relying exclusively on the PDF.

3. This work appears to focus on utilizing past information, possibly generated by applying interventions to an existing dataset. However, what is the advantage of introducing an additional corresponding past state (perhaps the original one before intervention)? Why not treat this simply as a standard forecasting problem? Although the authors claim that causality plays a role, this assertion is not sufficiently justified. Moreover, since one past state could result from different interventions applied to various factual cases, would incorporating such additional causal structures introduce greater complexity or instability compared to a conventional forecasting framework?

4. In Appendix B.5, the details on how the interventions were constructed are still missing. To demonstrate the practical value of this work, especially in real-world scenarios involving complex patterns, it is essential to justify that both the model design and the simulated data can effectively support training for complex pattern recognition. Unfortunately, the current version of the paper does not provide sufficient evidence to substantiate this key point.

5. The reproducibility of this work is another major concern. The paper lacks essential records of hyperparameter settings, which is particularly important given the inherent outcome uncertainty of the generative AI model employed. Furthermore, the details of how the simulation data were constructed and how the model was trained to adequately capture complex real-world patterns, so as to perform as claimed in the case study, are not clearly documented. These omissions make the reproducibility and reliability of the reported results questionable.

I would consider raising my scores if the above concerns are adequately addressed.

---

> ### Author Response · Authors · 2025-11-17
> **Reply to Reviewer dDVY**
>
> Thank you very much for your thoughtful review! We fully agree that the Introduction and Related Work sections in our initial submission lacked clarity. We have now *carefully rewritten Section 1 and Related Work parts*, directly addressing all of your suggestions. The updated PDF reflects these revisions. Below, we summarize how we responded to each of your concerns:
>
> > W1: Several key concepts are not clearly introduced at the beginning.
>
> We now clearly define both the interventional the counterfactual queries at the beginning of the Introduction, with detailed examples.
>
> We now clearly define the *discrete action variables* and the meaning of “extrapolating a counterfactual path and contrasting it” in the *Causal Effects on Time Series* subsection (Lines 97-107).
>
> We removed the earlier vague sentence: “these works do not encode causal DAGs or simulate system-wide trajectories under interventions”, and replaced it with precise distinctions from prior work (Lines 114–117).
>
> > W2: I would prefer the authors to directly explain why the research problem is important; The authors are also advised to briefly outline the key challenges.
>
> As suggested, we now explicitly explain the importance of both *interventional* and *counterfactual* reasoning through examples (Lines 38-53). We also clearly state what is missing in prior work and articulate the key motivations behind our algorithm (Lines 54–58).
>
> > W3: Organization of Contributions and Related Works:
>
> We moved the Contributions into the Introduction (Lines 59–65) and ensured that the Introduction now focuses solely on (i) defining the problem, (ii) explaining what is lacking in existing approaches, and (iii) motivating our method.
>
> We then introduce all related work in Section 1.1. The section begins with *Time-series forecasting* and *Causal generative modeling* (on static data), and then situates our work relative to two complementary causal research lines on time series: *Causal effects* and *Causal discovery*.
>
> > W3: Does the proposed method fall under an existing category of related work? How does it differ from similar studies?
>
> Our paper addresses a new research direction: *interventional and counterfactual reasoning in time series*. Existing causal time-series research falls into the following categories (summarized in Related Works):
> * Causal effects on time series
> * Causal discovery
>
> Our method is connected to Category 1: it provides interventional forecasts that can be used directly for causal-effect estimation. To highlight this connection, we added a **new real-world experiment on cancer treatment outcomes** (Section 5.3).
>
> Category 2 (causal discovery) aims to learn a causal DAG from observational data, whereas our method assumes a known DAG. These methods therefore can serve as **upstream tools** that supply the DAG needed by DoFlow, which we have discussed in *Limitations* of Section 5.4.
>
> We now clearly stated what’s lacking in causal time-series research and thus positioned our work in Lines 114-121.
>
>
> > W4: It is recommended that the authors introduce the concept of a causal DAG and explain more on how a multivariate time series can be represented as a directed graph;
>
> We now explicitly define the parent set pa(i) and how to construct the causal DAG in Lines 130–135. For instance, in physical systems, upstream components are permanent causal parents of downstream ones, and this structure is shared across all time steps. We also provide a concrete example of constructing the hydropower causal DAG in Lines 1540–1546.
>
> When the causal structure is unknown, we also clarify (Lines 514–519) that existing *causal discovery* methods can be applied as an upstream step to learn the DAG before applying DoFlow.
>
> > W5: Definitions of notations
>
> We have clarified all notation and definitions highlighted in this concern.
> - $\gamma$ is now defined in Lines 151-153.
> - We distinguish between $X$ (random variable) and $x$ (realized value) in Line 147.
> - We explain $H_{i,t}$ at its first appearance in Lines 164-169.
> - We make it explicit in (7) that $H_{i,t-1} := \mathrm{concat}\big(h_{i,t-1},\, h_{\mathrm{pa}(i),t-1}\big)$ is a concatenation of hidden-state vectors.

---

> ### Author Response · Authors · 2025-11-17
> **Reply to Reviewer dDVY (Second Part)**
>
> Below, we answer your Questions in addition to the presentation issues raised in Weaknesses. Thank you very much for raising these meaningful questions.
>
> *We summarize your Questions and give answers below. Please let us know if we didn't understand your questions correctly.*
>
> > Since DoFlow draws forecasts from a learned distribution, does this stochasticity reduce interpretability?
>
> No. For interventional forecasting, DoFlow models the *unobserved randomness* $U_{i,t}$ of the structural causal model (SCM): $X_{i,t} := f_i(X_{i,t-}, X_{\mathrm{pa}(i),t-}, U_{i,t})$. To predict time series value $x_{i,t}$, we sample $Z_{i,t}\sim N(0,1)$ and decode it to obtain $X_{i,t}$.
>
> We showed in Proposition 4.3 that there exists a continuously differentiable bijection $g$ such that $Z_{i,t} = g(U_{i,t})$. Therefore, the sampling variable $Z_{i,t}$ in DoFlow is a reparameterization of the true exogenous noise $U_{i,t}$ in the SCM.
>
> This means that DoFlow does not introduce additional artificial uncertainty; instead, it learns and reproduces the true randomness of the underlying causal process.
>
> > Can the model capture rare or extreme events, or will the PDF smooth them away?
>
> Yes, the model captures rare events. Because DoFlow conditions on the causal DAG and the hidden representation $H_{i,t}$, it learns the full conditional density, including tail behavior.
>
> When the context window of a time series is abnormal, the corresponding $H_{i,t}$ becomes abnormal, and the flow model generates future trajectories conditioned on this rare-event condition. This behavior is visible in **Figure 3**.
>
> However, our focus is on modeling the interventional and counterfactual distributions under the i.i.d. setting. While the model can represent rare events when they are supported in the observed data, systematically modeling rare events is **outside the scope** of this work. We have now clarified this in Lines 536–538.
>
> > Why do you introduce the recurrent state $H_{i,t}$? Why not treating our setting as a standard forecasting problem? How is causality concretely used?
>
> The recurrent hidden state $H_{i,t}$ summarizes the past of node $i$ and its parents $pa(i)$. Crucially, $H_{i,t}$ is always updated from the currently simulated trajectory (interventional or counterfactual). When we intervene with $X_{i,t} = \gamma_{i,t}$, this intervened value is fed into the RNN update, so $H_{i,t}$ **explicitly carries the interventional history** forward along the DAG.
>
> A standard forecaster learns $p(X_{\tau+1:T} | x_{1:\tau})$, and once $x_{1:\tau}$ is fixed, produces the same forecast regardless of hypothetical changes. We now have clearly explained interventional and counterfactual with detailed examples in Lines 38-53. Here we use **counterfactual** as a concrete illustration:
>
> *Counterfactual*: Consider a healthcare setting where we observe a patient’s treatment history and outcome trajectory, and then ask whether this particular patient’s trajectory would have been better or worse under a different dosing schedule. Outcomes depend not only on dosing but also on patient-specific factors that are *not directly observed* (e.g., baseline health), yet these are implicitly reflected in the factual trajectory. By conditioning on the observed factual trajectory, we can infer these unobserved factors, and then re-simulate how the same patient’s trajectory would have evolved under the alternative dosing plan.
>
> Algorithm 2 implements exactly this procedure. Using the forward process (Eq. (11)) with factual states $H_{i,t}^{F}$, we encode $z_{i,t}^{F}$, which captures the individual-specific unobserved factors inferred from the factual path. Then, using the reverse process (Eq. (12)) starting from $z_{i,t}^{F}$, but under counterfactual hidden states $\hat{H}_{i,t}^{CF}$ that carry the interventional history, we generate the counterfactual trajectory. Section 4 provides theoretical support for this counterfactual recovery. To our knowledge, DoFlow is the first to enable such counterfactual reasoning in time-series.
>
> > Since one past state could result from different interventions applied to various factual cases, would incorporating such additional causal structures introduce greater complexity or instability?
>
> The model is trained only on observational data, and it learns the functions $X_{i,t} := f_i(X_{i,t-}, X_{\mathrm{pa}(i),t-}, U_{i,t})$, rather than relying on pre-defined $f_i$. The causal DAG simply specifies the parent set for each node. This enforces the correct causal ordering and, as shown in Tables 1, 5, and 6, **improves observational forecasting** compared with baselines that do not encode a DAG.
>
> More importantly, the DAG is what makes interventional and counterfactual forecasting well-defined—capabilities that standard forecasters lack.
>
> DoFlow does assume a known causal DAG, and we added *Limitations* in Section 5.4, discussing the case where the DAG is unknown and extensions.

---

> ### Author Response · Authors · 2025-11-17
> **Reply to Reviewer dDVY (Third Part)**
>
> > The details on how the interventions were constructed are missing; The reproducibility of this work is not clearly specified.
>
> We have now clearly specified the interventional simulation procedure in **Appendix D.5**, and we have clearly specified the parameters in simulating the data in **Appendix D.1-D.4**. We added a new **Appendix E: Model and Implementation Details**, where we document all model configurations, hyperparameters, and training procedures. The newly included causal treatment–effect experiment (Section 5.4) has also been updated with its training details. Finally, we include the full code in the Supplementary Materials to ensure reproducibility.
>
>
> > Below, we summarize our **key contributions**:
>
> 1. *Introducing interventional and counterfactual forecasting to time series*:
>
> We bring interventional and counterfactual reasoning into time-series forecasting—an area where interventional forecasting has been under-explored (existing works focus mainly on treatment-effect estimation), and where counterfactual forecasting, to our knowledge, has not been explored in prior work.
>
> 2. *Theoretical guarantees for counterfactual recovery*:
>
> We establish formal counterfactual recovery properties of DoFlow in Section 4 under Assumption 4.1, showing that the model can correctly identify unobserved exogenous noise and recover exact counterfactual trajectories.
>
> 3. *Comprehensive empirical evaluation on synthetic and real-world data*:
>
> We conduct extensive synthetic experiments with quantitative metrics, evaluate observational and interventional forecasting on a real Hydropower dataset. Besides, new in the revision, we demonstrate DoFlow’s applicability to causal-effects problems through a real-world case study on cancer treatment outcomes (Section 5.4).
>
> We sincerely appreciate your valuable time and review. We look forward to any further comments you may have!

---

> ### Comment · Reviewer_dDVY · 2025-11-26
>
> Thank you for your further interpretation. Unfortunately, my core concern has still not been addressed.
>
> 1. My Question 2 was: “According to the mechanism of generative AI, particularly the flow-matching approach adopted in this work, the output represents only a sample from a potential distribution. Would this introduce additional uncertainty that complicates interpretability?” Your response — “This means that DoFlow does not introduce additional artificial uncertainty; instead, it faithfully learns and reproduces the true randomness of the underlying causal process.” — indeed acknowledges the randomness inherent in the process. However, this returns to my original concern: would this sampling-based nature introduce additional uncertainty that complicates interpretability? Here, the “additional uncertainty” I refer to is not the inherent stochasticity of the data-generating process itself, but rather the uncertainty imposed on interpretability, which is ideally expected to be stable and deterministic. **Such uncertainty renders both the results and the interpretability less convincing and less reproducible, due to the mechanism of generative AI methods such as diffusion or flow-matching.**
>
> 2. For my question, “Can the model capture rare or extreme events, or will the PDF smooth them away?'', your response --- “Yes, the model captures rare events. Because DoFlow conditions on the causal DAG and the hidden representation $H_{i,t}$, it learns the full conditional density, including tail behavior.'' --- primarily addresses the issue from a model architecture perspective. **However, for deep learning methods, performance depends not only on the model design but, more critically, on the training process and the data used to support it.** My original concern was that, based on the description of the training procedure, it is not convincing that the data synthesis with regular patterns is sufficient to effectively support the learning of complex patterns, particularly rare or extreme events. It is therefore essential to justify that both the architecture and the training data distribution jointly enable robust learning of tail behaviors. Unfortunately, the current version of the paper does not provide adequate empirical or methodological evidence to substantiate this key point. Consequently, I remain skeptical about the effectiveness and reproducibility of this work, especially with respect to its claimed ability to reliably model rare or extreme events under the current training conditions.

---

> ### Comment · Reviewer_dDVY · 2025-11-26
>
> 3. For "Appendix E: Model and Implementation Details'', the described process for DATA SYNTHESIS remains identical to the previous version, which was already unconvincing. In particular, the reliance on highly regular and structured synthetic patterns does not sufficiently demonstrate that the model can be effectively trained to capture the complex, irregular, and diverse anomaly patterns encountered in practical real-world scenarios. Without stronger evidence that the synthetic data adequately reflects the variability and richness of realistic anomalous behaviors, **it is difficult to be convinced that such regular patterns can support robust training for practical complex pattern recognition, further reinforcing concerns regarding the effectiveness and reproducibility of the proposed approach.**
>
> Overall, the work still appears to me as an attempt to address a complex real-world problem through technically fancy and narratively good-looking stories, without sufficient conceptual grounding or convincing empirical evidence to support its claims. The reproducibility, practical applicability, and generalizability of this work remain notably limited at this stage. As such, it risks misleading readers into investing time and resources in tedious and highly uncertain engineering-level parameter tuning while ending up with ineffective performance in practical complex scenarios.

---

> ### Author Response · Authors · 2025-11-26
> **Response to Reviewer dDVY Follow-up Comments [1/2]**
>
> Dear Reviewer dDVY,
>
> We appreciate your in-time response! Thanks for the detailed clarification, and we now better understand your concern. Please see our continued response, we sincerely appreciate your time:
>
> > 1. Such uncertainty renders both the results and the interpretability less convincing and less reproducible, due to the mechanism of generative AI methods such as diffusion or flow-matching.
>
> First, for *counterfactual forecasting*, we fully agree that interpretability, stability, and identifiability are important. In DoFlow, counterfactual trajectories are *not* generated via random sampling: once the factual sequence and intervention are given, it deterministically produces a **unique** counterfactual estimate. In Section 4, under Assumptions (A2) and (A3), we formally prove that this estimate is uniquely identifiable and corresponds to the true counterfactual outcome.
>
> Second, for *interventional* and *observational* forecasting, our goal is to model the full **conditional distribution** $p(X_{\tau+1:T}|x_{1:\tau},do(\cdot))$, because the underlying SCM itself is stochastic due to exogenous noise. In this case, **providing only a single deterministic point estimate is actually less interpretable**, as it hides the uncertainty inherent in the system.
>
> In DoFlow, this uncertainty is not artificially introduced by the generative mechanism. Proposition 4.3 shows that each sampled latent $Z_{i,t}\sim N(0,1)$ corresponds bijectively to an exogenous noise $U_{i,t}$ in the true data generating process. This means that sampling is to explore the true underlying distribution.
>
> On the contrary, deterministic approaches predict only the mean of the forecasting distribution and cannot quantify how confident we are in that estimate. They are often efficient and accurate, but provide no measure of reliability. DoFlow, as a probabilistic model, generates a set of samples, from which we compute prediction intervals—the narrower the interval, the higher the confidence—thus offering both the estimate and its uncertainty, rather than relying on a single point prediction.
>
> > 2. Consequently, I remain skeptical about... especially with respect to its claimed ability to reliably model rare or extreme events under the current training conditions.
>
> Thank you for highlighting this! We realize our previous response may have overstated the role of rare-event modeling, and we would like to clarify the scope of our claims.
>
> Our primary goal is **not** to “reliably model rare or extreme events under distribution shift,” but to introduce a framework for interventional and counterfactual time-series forecasting under the standard i.i.d. assumption, i.e., train and test come from the same SCM and intervention family. We do not claim guarantees for very heavy tails or out-of-support extremes, and we agree that addressing those would require additional assumptions and targeted experimental design.
>
> Our Section 3.3 (“Additional Property: Likelihood-based Anomaly Detection”) makes a much narrower claim: assuming the model is trained only on normal trajectories, when test inputs deviate from this distribution, DoFlow predicts significantly lower log-likelihood—thereby enabling anomaly detection, **not** reliable modeling of tail events.
>
> The rare event modeling is **outside the scope** of this work, and we have now stated this explicitly in Lines 536-538.
>
> > 3. (i) For "Appendix E: Model and Implementation Details'', the described process for DATA SYNTHESIS remains identical to the previous version.
>
> We thoroughly expanded the **data synthesis details in Appendix D, but not in Appendix E**, which focuses on model architecture and training details. We apologize for the confusion.
>
> In the rebuttal version, we have explicitly specified the choice of $\beta_i$ and the full simulation equations (Eq. 47–54), and Appendix D.5 now clearly explains how interventions are constructed for evaluation.
>
> *Most importantly, we have provided the test datasets and full code with our submission to fully ensure reproducibility.*
>
> >3. (ii) It is difficult to be convinced that such regular patterns can support robust training for practical complex pattern recognition.
>
> Thanks for raising this important point. The synthetic data are not limited to regular patterns. As detailed in Appendix D, we simulate four structurally different causal DAGs, each with both **linear additive** and **nonlinear, non-additive** mechanisms, including interaction and stochastic effects, to support complex pattern learning.
>
> For more reliable real-world applicability, beyond the Argonne hydropower case (Section 5.2), we added **new experiments Section 5.3: Cancer Treatment Outcomes**. We have made great efforts in showing the applicability across diverse scenarios.

---

> > ### Author Response · Authors · 2025-11-26
> > **Response to Reviewer dDVY Follow-up Comments [2/2]**
> >
> > > As such, it risks misleading readers into investing time and resources in tedious and highly uncertain engineering-level parameter tuning while ending up with ineffective performance in practical complex scenarios.
> >
> > We respectfully clarify that this work is **not** an exercise in parameter tuning or engineering-level optimization on specific datasets. Instead, it addresses a fundamental and under-explored problem: how to perform interventional and counterfactual forecasting in time-series governed by causal DAGs.
> >
> > We intentionally **did not rely on heavy parameter tuning**. As shown in Appendix E, all model hyperparameters and architecture settings are kept identical across synthetic datasets and real-world Argonne hydropower dataset:
> > - Per-node RNN encoder: 3-layer LSTM with hidden size 15.
> > - Per-node CNF: MLP with hidden dimension 64 and 3 layers; the conditioning dimension is: $15 + |pa(i)|\cdot 15$.
> >
> > Finally, our method does **not apply a fancy model to an existing well-studied problem**. For counterfactual forecasting, an encoder–decoder design is theoretically required to abduct exogenous latent noise from factual trajectories and reuse it under interventions.
> >
> > The forward-reverse processes of flow naturally provides a mechanism for this, and in time-series settings, requires conditioning on $H_{i,t}$ to capture past temporal and interventional context. This architectural choice is **driven by causal principles**, not engineering ornamentation.
> >
> >
> > Thank you once again for your time and thoughtful reviews, which have indeed helped us clarify and strengthen the entire paper! We look forward to any further comment you may have!

---

### Official Review · Reviewer_sw48 · 2025-10-31

**Soundness:** 4
**Presentation:** 3
**Contribution:** 3
**Rating:** 6
**Confidence:** 3

**Summary:**

This paper proposes **DoFlow**, a time-conditioned continuous normalizing flow (CNF) defined over a causal DAG for multivariate time series. Each node’s flow is conditioned on recurrent states of the node and its parents, enabling an encode–decode procedure that supports observational, interventional, and counterfactual forecasting via abduct–action–predict. The training uses a conditional flow-matching loss; the model also exposes trajectory log-likelihoods used for automatic change-point/anomaly detection. On the theory side, under assumptions (independent noise, monotonicity in the noise, and a base-law condition on the encoded latent), the authors show that the encoded latent is a bijective function of the exogenous noise and derive a counterfactual recovery result for the encode–decode scheme; they relate this to BGM, noting their result does not require observational distribution matching. Empirically, they evaluate on synthetic DAGs (tree/diamond/chain/FC-layer) for interventional and counterfactual rollouts, and on a real hydropower system, reporting interventional/observational RMSEs and demonstrating early outage detection from log-densities (often 10–20 minutes in advance).

**Strengths:**

* Encoder–decoder CNF over a causal DAG neatly instantiates **abduction–action–prediction**, so observational, interventional, and counterfactual forecasts all come from one coherent mechanism.
* **Impressive empirical results**: strong observational/interventional accuracy and compelling real-world demos (incl. early anomaly/outage detection), plus robustness across diverse synthetic DAGs.
* **Theory that matches the use case**: clear assumptions lead to bijectivity-in-noise and a **counterfactual recovery** result that directly supports the encoder–decoder design.
* **Clear, well-structured presentation** that makes the model easy to follow and reproduce.

**Weaknesses:**

* **Positioning vs prior causal–flow work could be clearer.** The paper does a good job situating the CNF side (Neural ODEs, flow matching, encode–decode) but is lighter on contrasting with prior *causal* modeling using flows (e.g., works like *Causal Normalizing Flows: From Theory to Practice*, Javaloy et al.). A short subsection disentangling what comes from causal literature, what from flows, and what’s **novel here** would help readers map contributions more precisely.
* **Prediction intervals lack calibration evidence.** Interventional figures show 50%/90% bands, but I couldn’t find quantitative **coverage** or calibration metrics (e.g., empirical coverage, interval width/ACE, Winkler score, CRPS) to support the claim. Consider reporting coverage on synthetic data (where ground truth is known).
* **Discussion section is a bit too sparse in my opinion** Even though the results seem good, I think a small discussion on particular on the limitations of this work would be interesting.

**Questions:**

* Following weakness 2, do you have any results for the calibration of the output intervals?
* Did you consider to replacing the RNN by a small transformer or just an attention layer? It would be more costly at prediction time, but would be interesting to see if it improves performance, and there could maybe even have some speed-ups in the case of a large past time series with a small forecast, due to parallelization.

---

> ### Author Response · Authors · 2025-11-17
> **Reply to Reviewer sw48**
>
> Thank you sincerely for your time and thoughtful review! We acknowledge that the Introduction and Related Works sections in the initial version lacked clarity. We have carefully rewritten the Introduction, Related Works, and Discussion sections, and would be very grateful if you could review the revised version. Below, we address your questions point by point and indicate where the corresponding revisions can be found in the paper:
>
> > Positioning vs prior causal–flow work could be clearer. A short subsection disentangling what comes from causal literature, what from flows, and what’s novel here would help readers map contributions more precisely.
>
> Based on your comments, we have reorganized and clarified the positioning in Section 1.1 (Related Works). We now begin by introducing modern *Time-series forecasting* methods (Lines 72–82) and *Causal generative modeling* on static data (Lines 83–91). We then present two research lines complementary to ours: *Causal effects on time series* and *Causal discovery*. After that, we clearly articulate what is missing in current causal time-series literature (Lines 114–121).
>
> To improve logical flow, we also moved the Contributions section to precede Section 1.1, and we added more recent works in each category to better situate our method within the existing literature.
>
> > Prediction intervals lack calibration evidence.
>
> Thank you very much for this point! Indeed, DoFlow provides probabilistic observational and interventional forecasts. To address calibration, we have now added results for the **Continuous Ranked Probability Score (CRPS) in Table 6**, alongside probabilistic forecasting baselines. We also include in Appendix D.6.3 the formal definition of CRPS and our empirical evaluation procedure.
>
> The CRPS results complement our existing metrics: RMSE (Table 1) and MMD (Table 5). Overall, the results show substantial performance improvements.
>
> Notably, performance on counterfactual forecasting can only be evaluated using RMSE, since DoFlow predicts a unique counterfactual trajectory that recovers the true one (as shown in Corollary 4.5).
>
> *To our knowledge, counterfactual forecasting has not been explored in prior work.*
>
> > Discussion section is a bit too sparse in my opinion:
>
> Thank you for raising this point. We have now expanded the Discussion by adding a **Limitations** paragraph at the end of Section 5.4. In summary, DoFlow assumes a known causal DAG. When the causal structure is unknown, we explain that one can apply *causal discovery* methods (as discussed in Section 1.1) as an upstream step before using our model.
>
> In addition, we now explicitly state the fundamental structural causal model (SCM) assumptions underlying DoFlow in **Appendix B.1** and reference this appendix at the start of Section 4 (Theoretical Properties). We also provide a detailed discussion of these assumptions in **Remark B.2**, particularly on the *unconfoundedness* assumption in Assumption B.1. We also connect to causal generative models on static data that address deconfounding, discussing that an important future direction is to extend DoFlow to handle deconfounded or partially latent temporal DAGs.
>
> We hope these additions make the Discussion more complete and informative!
>
> > Did you consider to replacing the RNN by a small transformer or just an attention layer? It would be more costly at prediction time, but would be interesting to see if it improves performance, and there could maybe even have some speed-ups in the case of a large past time series with a small forecast, due to parallelization.
>
> This is an excellent point. In designing DoFlow, our focus was on introducing causal DAG–structured training for time series, enabling *interventional* and *counterfactual* forecasting, which is under-explored in prior work. Within this framework, the choice of an RNN for each node is well aligned with how DoFlow factorizes the joint process. Replacing the RNN with a transformer or attention block is certainly possible, but the RNN is more natural for below reasons:
>
> 1. The autoregressive causal mechanism of RNN matches the SCM structure $X_{i,t} := f_i(X_{i,t-}, X_{pa(i),t-}, U_{i,t})$, which is inherently recursive. By contrast, a transformer recomputes attention over the entire history at each step, which does not naturally match the recursive structure.
>
> 2. Interventions at arbitrary times require Markovian hidden-state propagation. Once an intervention changes a node at time $t$, the model must propagate its effect forward through the hidden state—exactly the Markovian update performed by an RNN.
>
> 3. RNNs are more parameter-efficient per node. DoFlow trains one small generative model and one encoder per node in the DAG. Using a RNN encoder for each node would be substantially more expensive compared to a transformer.

---

> > ### Author Response · Authors · 2025-11-17
> > **Reply to Reviewer sw48 (Second Part)**
> >
> > We also would like to briefly summarize how we addressed other reviewers’ concerns and improved the paper in the latest revision. We sincerely thank all the reviewers for your effort!
> >
> > 1. In response to Reviewer dDVY: We thoroughly rewrote the Introduction and Related Work sections to clarify the problem setup and positioning. We clarified the causal graph construction and notation in Section 2, and substantially expanded Appendices D and E to fully include experimental details (synthetic data generation, interventions, model configurations, and hyperparameters).
> >
> > 2. In response to Reviewer jniY: We clarified the overall setup and comparisons to modern causal and flow-based works in the Introduction. We strengthened Assumption 4.1 in Section 4 and added Assumption B.1 on SCM assumptions in Appendix B.1. We further expanded Section 5.4 and Appendix B to discuss model limitations (e.g., known DAG, unconfoundedness) and outline future directions.
> >
> > 3. In response to Reviewer Pjb9: We added concrete examples in the Introduction illustrating the practical value of counterfactual queries. We clarified the role of exogenous noise and the SCM in our counterfactual formulation in the rebuttal text. We added Appendix E with detailed experimental and implementation settings, and Appendix F.4 to clarify the likelihood-based anomaly detection procedure.
> >
> >
> > > Below, we also wish to summarize our **key contributions** of the paper. Thank you so much for reading!
> >
> > 1. *Introducing interventional and counterfactual forecasting to time series*:
> >
> > We bring interventional and counterfactual reasoning into time-series forecasting—an area where interventional forecasting has been under-explored (existing works focus mainly on treatment-effect estimation), and where counterfactual forecasting, to our knowledge, has not been explored in prior work.
> >
> > 2. *Theoretical guarantees for counterfactual recovery*:
> >
> > Beyond introducing counterfactual forecasting in time-series, we establish theoretical guarantees in Section 4 showing that DoFlow can recover the true counterfactual trajectory (Corollary 4.5) under Assumption 4.1.
> >
> > 3. *Comprehensive empirical evaluation on synthetic and real-world data*:
> >
> > We conduct extensive synthetic experiments with quantitative metrics, evaluate observational and interventional forecasting on a real Hydropower dataset. Besides, new in the revision, we demonstrate DoFlow’s applicability to causal effects through a **new real-world experiment** on *cancer treatment outcomes* (Section 5.3).
> >
> > We sincerely appreciate your valuable time and feedback! We look forward to any further comments you may have!

---

### Meta-Review · Area_Chair_AK5h · 2026-01-09

**Summary:**

The reviewers overall were positive about the topic and potential contributions of the paper. However, they raised important concerns regarding i) the clarity of the motivation to and description of the problem; ii) completeness of the theoretical assumptions; iii) completeness of related work; and iv) the limited empirical evaluation and clarity of some of the results. As acknowledged by the reviewers, most of these concerns have been successfully addressed, improving the assessment of several of them.  I do personally agree that indeed the current version of the paper is above the acceptance bar.

Yet, I would encourage the authors to cite the related works [1,2] (even if not completely aligned with their approach), as otherwise the related work seems incomplete. Additionally, I believe the authors should also explicitly state those works that are the foundation to their theoretical results.  As far as I can see, the assumptions and proofs in this paper resemble significantly those in, e.g., [3]. Thus, I believe the authors should also position their work in the literature also from a theoretical perspective, even if here the results are generalized to the temporal setting.

[1] Liu, Y., Sun, Y., & Lim, J. H. (2023, June). Counterfactual Dynamics Forecasting–a New Setting of Quantitative Reasoning. In Proceedings of the AAAI Conference on Artificial Intelligence (Vol. 37, No. 2, pp. 1764-1771).

[2] Cinquini, Martina, et al. "A Practical Approach to Causal Inference over Time." Proceedings of the AAAI Conference on Artificial Intelligence. Vol. 39. No. 14. 2025.

[3] Javaloy, A., et al. (2023). Causal normalizing flows: from theory to practice. Advances in Neural Information Processing Systems, 36, 58833-58864.

**Reviewer Concerns:**

The reviewers acknowledge the novelty and potential impact of the paper, as well as the significant improvement that the authors have made on the paper during the rebuttal period. 3 our of the 4 reviewers are finally positive about the contributions of the paper.  The remaining reviewer (dDVY) raises important practical limitations. However, I believe that the contributions of the paper are still significant and thus that it will trigger future work addressing these practical limitations.

**Reviewer Scores:**

Some of the reviewers actually explicitly state so, leaving the paper with 3 positive reviewers. While the 4th reviewer does not seem as positive, I believe their remaining concerns are out of the scope of the paper, and that is why I am waiting this assessment a bit less than the others.

---

### Decision · Program_Chairs · 2026-01-26

Accept (Poster)